# SegVol: Universal and Interactive Volumetric Medical Image Segmentation

**Yuxin Du**[1,2]**, Fan Bai**[1,2,3]**, Tiejun Huang**[2,4]**, Bo Zhao**[1,2†]
[1]School of Artificial Intelligence, Shanghai Jiao Tong University
[2]BAAI   [3]The Chinese University of Hong Kong   [4]Peking University
[†]Corresponding author: Bo Zhao <bo.zhao@sjtu.edu.cn>

## Abstract

Precise image segmentation provides clinical study with instructive information. Despite the remarkable progress achieved in medical image segmentation, there is still an absence of a 3D foundation segmentation model that can segment a wide range of anatomical categories with easy user interaction. In this paper, we propose a 3D foundation segmentation model, named *SegVol*, supporting universal and interactive volumetric medical image segmentation. By scaling up training data to 90K unlabeled Computed Tomography (CT) volumes and 6K labeled CT volumes, this foundation model supports the segmentation of over 200 anatomical categories using semantic and spatial prompts. To facilitate efficient and precise inference on volumetric images, we design a zoom-out-zoom-in mechanism. Extensive experiments on 22 anatomical segmentation tasks verify that SegVol outperforms the competitors in 19 tasks, with improvements up to 37.24% compared to the runner-up methods. We demonstrate the effectiveness and importance of specific designs by ablation study. We expect this foundation model can promote the development of volumetric medical image analysis. The model and code are publicly available at: https://github.com/BAAI-DCAI/SegVol.

## 1 Introduction

Volumetric medical segmentation, involving extracting 3D regions of interest, such as organs, lesions, and tissues, plays a pivotal role in medical image analysis by accurately modeling the 3D structural information of the human body from volumetric medical images such as CT or MRI. The accurate segmentation can benefit numerous clinical applications including tumors monitoring[1, 2], surgical planning[3, 4], disease diagnosis[5], therapy optimization[6, 7], etc.

Compared to 2D medical image segmentation[8, 9, 10, 11, 12, 13, 14, 15, 16, 17], volumetric image segmentation is notably more challenging due to the labor-intensive annotation and resource-consuming computation. Recently, the research of volumetric medical image segmentation has garnered substantial attention, leading to a series of advancements[18, 19, 20, 21, 22, 23]. However, existing volumetric medical segmentation methods have several key limitations which prevent their application in challenging tasks, e.g., liver tumor or colon cancer segmentation[24, 25, 26, 27], and real-world tasks, e.g., human-interactive segmentation[28, 29, 30, 31, 32].

Firstly, the publicly available volumetric medical image datasets usually consist of a small number of mask annotations from a few varying categories. Due to the different label spaces, the traditional task-specific segmentation models trained on one dataset have difficulty in generalizing to others. For example, the CT-ORG dataset[33, 34, 24, 35] contains the 'lungs' category, while this category is split into two sub-classes and named 'left lung' and 'right lung' in the LUNA16 dataset[36]. Hence, a universal segmentation model has to understand the semantics of anatomical categories. Secondly,

38th Conference on Neural Information Processing Systems (NeurIPS 2024).

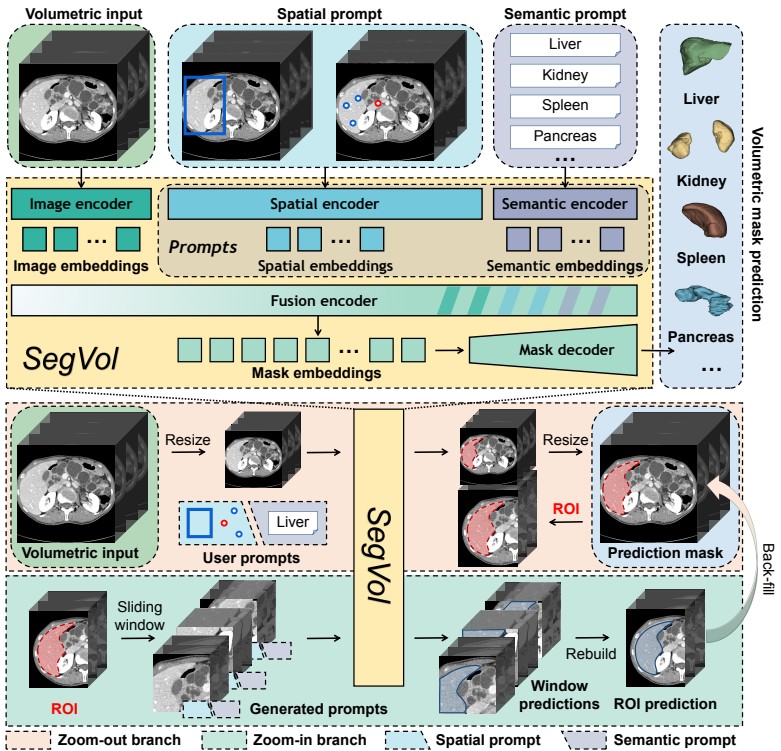

Figure 1: Overview of SegVol model architecture. SegVol produces precise segmentation of 3D anatomical structures from volumetric inputs with easy user interactions, including point, bounding box, and text prompts. Zoom-out-zoom-in mechanism: SegVol initially produces a rough prediction mask with zoom-out inference, then refines it with zoom-in inference on the identified ROI.

traditional segmentation models have inferior performance when segmenting complex structures, such as tumors and cysts[37]. This is because these models are trained on insufficient data and are also not able to leverage the spatial information through user interaction. Thirdly, previous solutions are computationally expensive in the inference process. They typically employ a sliding window to infer the whole volumetric input. This strategy is not only time-consuming but also short-sighted, as the sliding window contains only local information. Recently, there have been some works[29, 38, 39] that introduce spatial-prompt into medical image segmentation, shown in Table 1. However, most of them lack the ability to process the 3D input directly and naturally, and none of them is able to understand the semantics of anatomical categories.

In this paper, we propose the first foundation model for volumetric medical image segmentation – *SegVol*. The proposed model enables universal and interactive 3D segmentation of more than 200 anatomical categories, supporting both spatial and semantic prompts. SegVol can also be driven by the combination of multi-prompt, like 'bounding box+text' or 'point+text' prompts, achieving high-precision segmentation and semantic disambiguation. To enable efficient and precise segmentation of volumetric images, we develop a zoom-out-zoom-in mechanism that enables the model to be efficient and precise. We evaluate the proposed SegVol on 22 volumetric medical image segmentation tasks and the results demonstrate our method surpasses other SAM-like interactive segmentation methods[28, 38, 39, 29] by a large margin. Extensive case studies and ablation experiments are also carried out to prove the advantages of SegVol and the effectiveness of the zoom-out-zoom-in mechanism and multi-prompt combination.

We summarize our key contributions as follows:

1. Collect and process 25 public volumetric medical segmentation datasets, encompassing over 200 anatomical categories. The pseudo label is introduced to relieve the spurious correlation in the training data.

Table 1: The different settings and functions of SAM-like interactive segmentation methods.

| Method | Image Domain | Dimension | Training | Point | Bbox | Text | Inference Input |
|--------|--------------|-----------|----------|-------|------|------|-----------------|
| | | | | Prompt Type | | | |
| SAM[28] | Natural | 2D | Full-Param | ✓ | ✓ | ✓ | 1024×1024 |
| MedSAM[29] | Medical | 2D | Decoder | ✗ | ✓ | ✗ | 1024×1024 |
| SAM-Med2D[38] | Medical | 2D | Adapter | ✓ | ✓ | ✗ | 1024×1024 |
| SAM-Med3D[39] | Medical | 3D | Full-Param | ✓ | ✗ | ✗ | 128×128×128 |
| **OURS** | **Medical** | **3D** | **Full-Param** | ✓ | ✓ | ✓ | **Full Resolution** |

2. Implement massive 3D pre-training on 96K CT volumes and supervised fine-tuning on the 6k labeled datasets.

3. Support spatial-prompt, semantic-prompt, and combined-prompt segmentation, achieving high-precision segmentation and semantic disambiguation.

4. Design a zoom-out-zoom-in mechanism that significantly reduces the computational cost, meanwhile preserving precise segmentation.

## 2 Methodology

### 2.1 Dataset Construction

One of the main challenges of training a universal volumetric medical segmentation model is the absence of large-scale publicly available volumetric medical data, especially CTs with segmentation annotations. Doing our utmost, we collected 25 open-source segmentation CT datasets, including CHAOS[40, 41, 42], HaN-Seg[43], AMOS22[44], AbdomenCT-1k[45], KiTS23[46], KiPA22[47, 48, 49, 50], KiTS19[51], BTCV[52], Pancreas-CT[53, 54, 35], 3D-IRCADB[55], FLARE22[56, 57], TotalSegmentator[58], CT-ORG[33, 34, 24, 35], VerSe19, VerSe20[59, 60, 61], SLIVER07[62], QUBIQ[63], six MSD datasets[56], LUNA16[36], and WORD[64]. Their detailed information and availability are shown in the Section A. These CTs originate from various medical institutions, captured by different machines with varying parameter settings and scanning regions. To standardize these datasets, we use the mean voxel value of each volume to filter the background and then perform normalization on the foreground voxels.

Volumetric segmentation datasets suffer from the notorious problem of partial labels. Most of these datasets have annotations of only a few segmentation targets, e.g., several organs. Therefore, the deep models may learn the spurious correlation between datasets and segmentation targets, and thus produce inferior results during the inference phase. To relieve this problem, we introduce the pseudo labels by utilizing the Felzenswalb-Huttenlocher (FH)[65] algorithm to generate pseudo masks for each CT scan. Pseudo masks can supplement unlabeled categories in a dataset, therefore relieving the spurious correlation problem. To restrain the noise and numerous tiny masks in pseudo labels, we employ the following strategies: 1) The pseudo masks are replaced with ground truth masks when applicable. 2) We filter out tiny structures smaller than 1‰ of the whole volume size. 3) Each mask is refined by dilation and erosion operations.

### 2.2 Model Architecture

Motivated by the recent advance in 2D nature image segmentation, Segment Anything (SAM)[28], we design a novel model for interactive and universal volumetric medical image segmentation, named, SegVol. The model is illustrated in Figure 1. SegVol supports three types of prompts for interactive segmentation: 'bounding box(bbox)' prompt, including the coordinates of two diagonal vertices; 'point' prompt, composed of a set of positive and negative points; and 'text' prompt, such as 'liver' or 'cervical spine C2'. The model consists of four modules: image encoder, text encoder, prompt encoder, and mask decoder.

We employ 3D ViT (Vision Transformer)[66, 67] as the image encoder, which exhibits remarkable advantages over convolutional models[68] when pre-trained on large-scale datasets. The 3D ViT structure is designed as follows: patch size=(4, 16, 16), layers number=12, heads number=12, hidden

size=768. We first pre-train 3D ViT using SimMIM algorithm[69] on the collected 96K CTs, and then conduct further supervised fine-tuning on the 6K CTs with 150K labeled segmentation masks.

One of the main limitations of traditional segmentation models is that the models learn dataset-specific labels encoded as integers which cannot generalized to unseen datasets or tasks, preventing their real-world applications. We enable universal segmentation across datasets by leveraging the text encoder from CLIP model[70] to encode the input text prompt, as CLIP[70] has been trained to align image and text embeddings on web-scale image-text pairs. Given a word or phrase as the text prompt, we complete it using the template '*A computerized tomography of a [text prompt]*'[71] and then encode it into text embedding. The off-the-shelf text encoder is frozen during training due to the limited text data in CT datasets. Following SAM[28], we obtain the spatial-prompt embedding using positional encoding[72] on point and bbox prompt.

After obtaining the image embedding and prompt embedding, we input them into the mask decoder and predict the mask. We use self-attention and cross-attention in two directions to fuse the image embedding and prompt embedding, and then employ the transposed convolutions and interpolation operations to generate masks. Since text embedding is the key to universal segmentation and it is also challenging to learn the correlation between text and volumetric regions, we enhance the text information by introducing a parallel text input branch beside the joint prompt embedding.

### 2.3 Prompt Generation

SegVol accepts multiple types of prompts, including individual point, bbox, and text prompts, and also their combinations. To make full use of the segmentation training data, we generate kinds of prompts for each datum and construct kinds of prompt-mask data pairs for training.

The point prompt is built from ground truth or pseudo masks, consisting of three kinds of points, namely, positive point, negative point, and ignored point. Positive point means that it is within the target mask region, while negative points are those outside. Ignored points are utilized to ensure a uniform length of the point prompts for input completion. Notably, these ignored points are not considered by the model.

The bbox prompt is generated based on the ground truth or pseudo masks, integrated with random jitter to enhance the model's robustness. When generating the bbox prompt for some pseudo mask, the bbox may also cover other masks due to the irregular 3D shapes. To address this problem, we compute the Intersection over Union (IoU) between the generated bbox and the included pseudo masks. Any mask with an IoU greater than 0.9 will also be integrated and considered as part of the target mask corresponding to this bbox prompt.

The text prompts are constructed based on their category names. As pseudo masks produced by the unsupervised FH algorithm[65] do not have the semantic information, we only use point and bbox prompts for training on masks of pseudo labels.

### 2.4 Zoom-out-zoom-in Mechanism

SAM-like interaction with large-volume images is laborious for users, especially in the scene where the sliding window has to be used due to the limited view of those 3D models. To provide users with an easy SAM-like interface, we design a zoom-out-zoom-in mechanism, which is efficient and precise, consisting of zoom-out-zoom-in inference and multi-size training. As demonstrated in Figure 1, the zoom-out process involves resizing a volumetric image, which is input into the model with user prompts to generate a coarse segmentation mask. Then, the Region of Interest (ROI) from the original image is cropped for zoom-in analysis. In the zoom-in process, a sliding window is used to perform precise inference driven by prompts generated from the coarse segmentation mask. After that, the ROI prediction mask will be back-filled to the coarse segmentation mask to finish the final prediction. Besides, multi-size training involves augmenting the input data by resizing CTs for the zoom-out view and cropping them into cubes for the zoom-in view. The zoom-out-zoom-in mechanism realizes the computational cost reduction meanwhile producing precise segmentation of the ROI.

## 2.5 Loss Function

We apply SimMIM algorithm[69] to pre-train the image encoder of SegVol with the masked image modeling loss $\mathcal{L}_{\text{pre-training}}(\boldsymbol{\theta}_{\text{IE}}; \mathcal{D}_1)$. The loss function is as follows:

$$\mathcal{L}_{\text{pre-training}}(\boldsymbol{\theta}_{\text{IE}}; \mathcal{D}_1) = \frac{1}{\Omega(\boldsymbol{a}_{\text{M}})} ||\boldsymbol{b}_{\text{M}} - \boldsymbol{a}_{\text{M}}||_1, \tag{1}$$

where $\boldsymbol{\theta}_{\text{IE}}$ is the parameter set of SegVol's image encoder. $\boldsymbol{a}, \boldsymbol{b} \in \mathbb{R}^{D \times H \times W}$ are the input voxel values and predicted values, respectively. M denotes the set of masked voxels, $\Omega(\cdot)$ is the number of elements, and $\mathcal{D}_1$ is the pre-training dataset.

We combine the Binary Cross-Entropy (BCE) loss and Dice loss as the supervised fine-tuning loss function $\mathcal{L}_{\text{fine-tuning}}(\boldsymbol{\theta}; \mathcal{D}_2)$ to train the model with trainable parameters $\boldsymbol{\theta}$ (text encoder frozen). $\mathcal{D}_2$ is the supervised fine-tuning dataset and $\boldsymbol{x}, \boldsymbol{y} \in \mathbb{R}^{D \times H \times W}$ are the predicted mask and ground-truth mask, respectively. $\mathcal{F}(\cdot, \boldsymbol{\theta})$ is the forward function of SegVol. The loss function is as follows:

$$\mathcal{L}_{\text{BCE}}(\boldsymbol{\theta}; \mathcal{D}_2) = -\mathbb{E}_{(\boldsymbol{x},\boldsymbol{y}) \sim \mathcal{D}_2}[\langle \boldsymbol{y}, \log(\mathcal{F}(\boldsymbol{x}, \boldsymbol{\theta})) \rangle + \langle 1 - \boldsymbol{y}, \log(1 - \mathcal{F}(\boldsymbol{x}, \boldsymbol{\theta})) \rangle] \tag{2}$$

$$\mathcal{L}_{\text{Dice}}(\boldsymbol{\theta}; \mathcal{D}_2) = 1 - \mathbb{E}_{(\boldsymbol{x},\boldsymbol{y}) \sim \mathcal{D}_2}\left[\frac{2 \cdot \langle \boldsymbol{y}, \mathcal{F}(\boldsymbol{x}, \boldsymbol{\theta}) \rangle}{\|\boldsymbol{y}\|_1 + \|\mathcal{F}(\boldsymbol{x}, \boldsymbol{\theta})\|_1}\right] \tag{3}$$

$$\mathcal{L}_{\text{fine-tuning}}(\boldsymbol{\theta}; \mathcal{D}_2) = \mathcal{L}_{\text{BCE}}(\boldsymbol{\theta}; \mathcal{D}_2) + \mathcal{L}_{\text{Dice}}(\boldsymbol{\theta}; \mathcal{D}_2) \tag{4}$$

The detailed fine-tuning algorithm of SegVol is presented in Section B.

## 3 Experiments

In this section, we conduct extensive experiments on 22 volumetric medical image segmentation tasks to compare SegVol with other SAM-like medical image segmentation methods[28, 38, 39, 29]. Ablation studies are also carried out to prove the effectiveness of the zoom-out-zoom-in mechanism and provide more insights about dataset scale and multi-prompt combination. Detailed case studies are conducted to discuss the disambiguation ability of semantic-prompt and the capability of identifying the segmentation results with spatial-prompt.

### 3.1 Experimental Setup

During the pre-training, we follow SimMIM algorithm[69] to train the 3D ViT encoder of SegVol on the collected 96K CTs for 2000 epochs. In the supervised fine-tuning stage, we train SegVol (with the text encoder frozen) on the labeled 25 volumetric medical image segmentation datasets for 270 epochs with batch size 32 and input size (32, 256, 256), using AdamW optimizer[73]. SimMIM pre-training takes about $20 \times 8$ GPU hours, while fine-tuning takes about $300 \times 8$ GPU hours. All the above training process is implemented on 8 NVIDIA A100-SXM4-40GB. Three external datasets[44, 74, 75] and 20% testing data preserved from 25 collected datasets are used in the following experiments.

### 3.2 Compared with SAM-like Interactive Methods

Several efforts have been made to construct a SAM-like interactive medical image segmentation model. However, some of these works, such as MedSAM[29] and SAM-MED2D[38], focus on 2D tasks and cannot process 3D input directly. The other 3D-based methods, such as SAM-MED3D[39], only support small cropped input and do not support semantic-prompt segmentation, which are still far from building a comprehensive foundation model for volumetric medical image analysis.

**Competitors and configures.** In this experiment, MedSAM[29] and SAM(bounding box)[28] use bounding box prompts. SAM(5 clicks)[28], SAM-MED2D[38] and SAM-MED3D[39] use point prompts and a five-step correction procedure, which means that the point prompt in each step will be given according to the previous-step output and ground truth, rather than giving all at once. In

Table 2: Quantitative comparative experiment results for SegVol and other 5 SAM-like interactive segmentation methods settings in terms of the median value of Dice score.

| Dataset | Category | SAM(Point) [28] | SAM(Bbox) [28] | SAM-MED2D [38] | SAM-MED3D [39] | MedSAM [29] | OURS |
|---|---|---|---|---|---|---|---|
| AMOS22 [44] | Aorta | 0.7267 | 0.4362 | 0.8704 | 0.8102 | 0.3387 | **0.9273** |
| | Bladder | 0.4162 | 0.6281 | 0.8417 | 0.4338 | 0.6799 | **0.9120** |
| | Duodenum | 0.1554 | 0.3192 | 0.5066 | 0.3820 | 0.3066 | **0.7402** |
| | Esophagus | 0.2917 | 0.3541 | 0.5500 | 0.5174 | 0.3610 | **0.7460** |
| | Gallbladder | 0.2831 | 0.6161 | 0.7999 | 0.5643 | 0.6609 | **0.8763** |
| | Adrenal gland(L) | 0.0555 | 0.4222 | 0.5068 | 0.4584 | 0.3766 | **0.7295** |
| | Left kidney | 0.8405 | 0.8274 | 0.9325 | 0.8723 | 0.7909 | **0.9489** |
| | Liver | 0.7477 | 0.5124 | 0.6904 | 0.8801 | 0.6137 | **0.9641** |
| | Pancreas | 0.2127 | 0.3392 | 0.5656 | 0.5391 | 0.3217 | **0.8295** |
| | Postcava | 0.2042 | 0.5251 | 0.4436 | 0.6683 | 0.5211 | **0.8384** |
| | Prostate uterus | 0.2344 | 0.6986 | 0.7518 | 0.6231 | 0.7739 | **0.8557** |
| | Adrenal gland(R) | 0.0452 | 0.3642 | 0.1681 | 0.3708 | 0.3855 | **0.6994** |
| | Right kidney | 0.8459 | 0.8215 | 0.9077 | 0.8632 | 0.7851 | **0.9505** |
| | Spleen | 0.5936 | 0.6536 | 0.9267 | 0.8591 | 0.7038 | **0.9589** |
| | Stomach | 0.4229 | 0.3883 | 0.5399 | 0.4576 | 0.4378 | **0.9123** |
| | **Average** | 0.4050 | 0.5271 | 0.6668 | 0.6200 | 0.5371 | **0.8593** |
| ULS23 [74] | DeepLesion3D | 0.3686 | 0.7473 | 0.3258 | 0.2386 | **0.7680** | 0.7065 |
| | BoneLesion | 0.4461 | 0.6671 | 0.1947 | 0.4447 | 0.6896 | **0.6920** |
| | PancreasLesion | 0.0675 | 0.5579 | 0.5548 | 0.5526 | 0.6561 | **0.7265** |
| | **Average** | 0.2941 | 0.6574 | 0.3584 | 0.4120 | **0.7046** | **0.7046** |
| SegTHOR [75] | Aorta | 0.2744 | 0.3894 | 0.8077 | 0.7703 | 0.3278 | **0.8439** |
| | Esophagus | 0.0348 | 0.2046 | 0.3578 | 0.6394 | 0.2196 | **0.7201** |
| | Heart | 0.6695 | 0.8876 | 0.6012 | 0.8325 | **0.8924** | 0.8172 |
| | Trachea | **0.9147** | 0.1611 | 0.8306 | 0.8485 | 0.1261 | 0.8807 |
| | **Average** | 0.4734 | 0.4107 | 0.6493 | 0.7727 | 0.3915 | **0.8155** |

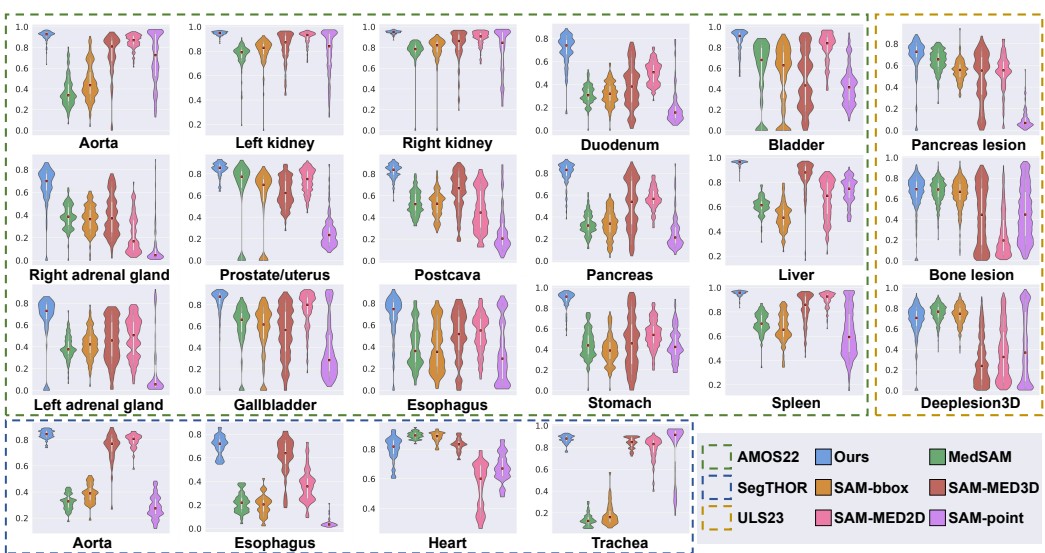

Figure 2: Violin plots for quantitative comparison experiment results of SegVol and SAM-like interactive methods[28, 38, 39, 29]. The vertical axis represents the Dice score.

this experiment, SegVol uses bounding box and text prompt which performs better than other kinds of prompt combinations. Detailed ablation study on prompt combination is demonstrated in Figure 3 (b). In addition, we compare SegVol with traditional task-specific segmentation models, e.g., 3DUX-NET[23], SwinUNETR[20], and nnU-Net[22], in Section C, though the direct comparison is unsuitable due to the different settings and objectives.

**Testing data.**  To compare with these SAM-like interactive segmentation models, we evaluate the models on 1,778 cases from the validation set of AMOS22[44], the whole novel annotated set of Uni-

Table 3: Ablation experiment on the zoom-out-zoom-in mechanism.

| Mechanism | Dice Score Avg. ↑ | Time Per Case Avg. ↓ |
|---|---|---|
| Resize | 0.4509 | 65 ms |
| Sliding window | 0.6529 | 3331 ms |
| **Zoom-out-zoom-in** | 0.7298 | 190 ms |

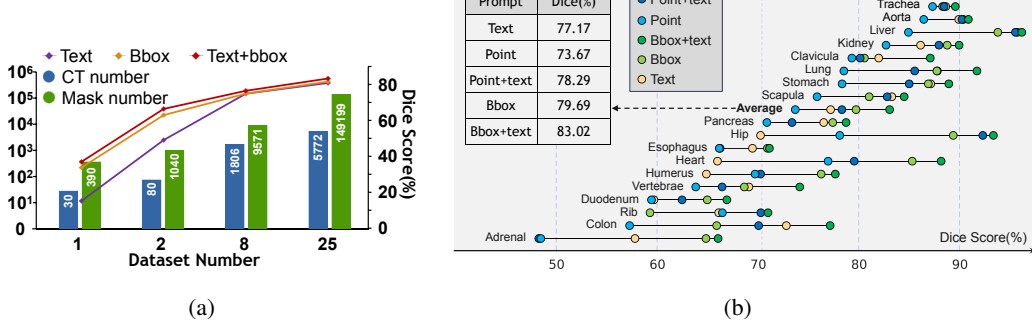

|  (a)  |  (b)  |

Figure 3: (a) The performance of SegVol improves as the training data scales up. (b) The quantitative experimental results on 19 anatomical segmentation tasks of split 20% test data demonstrate that using the combination of semantic and spatial prompts can achieve better performances.

versal Lesion Segmentation Challenge 23(ULS23)[74], and the released labeled set of SegTHOR[75]. The validation set of AMOS22 contains 120 cases annotated with 15 major organs. The novel annotated ULS23 dataset is composed of three subsets, namely, DeepLesion3D, Radboudumc Bone, and Radboudumc Pancreas. The DeepLesion3D subset contains 200 abdominal lesions, 100 bone lesions, 50 kidney lesions, 50 liver lesions, 100 lung lesions, 100 mediastinal lesions, and 150 assorted lesions cases. There are 744 bone lesion cases in the Radboudumc Bone subset and 124 pancreas lesion cases in the Radboudumc Pancreas subset. The 40 cases from SegTHOR, which are contoured manually by an experienced radiotherapist, focus on the heart, trachea, aorta, and esophagus that surround the tumor and must be preserved from irradiations during radiotherapy.

**Quantitative results.**   The quantitative results of comparative experiments are shown in Table 2, which verify our method is the best in most of the tasks including both lesions and organs, compared to other SAM-like interactive models[28, 38, 39, 29]. Specifically, our method outperforms the second-ranked SAM-MED2D on the AMOS22 dataset by a significant improvement of 19.25% (average Dice score). On the SegTHOR dataset, our method surpasses the runner-up – SAM-MED3D by an average Dice score improvement of 4.28%. The ULS23 dataset, characterized by small patch-like masks, presents a unique challenge. In this scenario, SegVol still exhibits good performance, comparable to MedSAM, which excels in using bbox prompts for segmenting small objects. We visualize the Dice score distributions of all methods in all the tasks as violin plots, depicted in Figure 2. More detailed results and visualization are present in Section C.

### 3.3   Ablation Studies

**Zoom-out-zoom-in mechanism.**   One of the key designs of SegVol is the zoom-out-zoom-in mechanism. We compare it with the intuitive resize strategy and the popular sliding window algorithm on the split 20% test data, 48 cases covering 15 major organs with a variety of sizes, belonging to the AMOS22[44] dataset. Two evaluation dimensions, i.e., performance (Dice score) and inference time cost (per case), are compared, as shown in Table 3. The zoom-out-zoom-in mechanism achieves the best average Dice score and a very competitive inference speed compared to the simple resize strategy. The reason for computational cost reduction is that the traditional sliding window method requires scanning the entire 3D CT and processing thousands of windows. In contrast, the proposed zoom-out-zoom-in mechanism only requires one global inference of 3D CT and then scanning the ROI with dozens of windows. Detailed experiment results are shown in Section C.

**Scaling up training data.** The success of scaling up training data has been witnessed in multiple computer vision tasks [28, 70]. We conduct an ablation study to investigate the importance of scaling up training images and masks. The split 20% test data of BTCV dataset[52], which includes 13 main organs, is set as an anchor to evaluate the model trained separately on 1, 2, and 8 datasets for 500 epochs, as well as the final model trained on 25 datasets. The detailed results are shown in Figure 3 (a). As a lightweight model, the performance of SegVol is weak when only one dataset is used. However, with the increase of training data, the Dice score increases rapidly, especially in the text prompt setting. The results indicate that our method is scalable and better performance can be achieved if more training data is available.

**Multi-prompt combination.** As a universal model, our approach achieves precise segmentation for over 200 organs, tissues, and lesions using both spatial and semantic prompts. In Figure 3 (b), we quantitatively analyze the mutually supportive relationship between semantic-prompt and spatial-prompt in 19 segmentation tasks of the 20% split test data. On the one hand, spatial-prompt allows the model to locate the specific part in the 3D space. According to Figure 3 (b), the average Dice score of the 'bbox+text' prompt is boosted by 5.85% compared to the 'text' prompt on average. On the other hand, semantic-prompt clarifies the reference to the anatomical structure, eliminating the ambiguity of spatial-prompt and the plausible masks of multiple categories. This is reflected in Figure 3 (b) as the average Dice score of 'point+text' prompts is 4.62% higher than using 'point' prompts alone. Spatial and semantic prompts mutually support each other, ultimately endowing the model with powerful segmentation capabilities.

## 3.4 Case Studies

**Disambiguation via semantic-prompt.** It is a notorious problem in interactive segmentation that one spatial-prompt may correspond to multiple plausible outputs [28]. As illustrated in the images on the top left in Figure 4, three of them correspond to three anatomical concepts, namely, kidney tumor, left kidney, and the whole kidneys, while they are all plausible to the same point prompt. Similarly, in the bottom left three images, the bounding box selects the region of the liver. However, liver tumors, hepatic vessels, and the liver itself are also plausible target structures. In these cases, SAM chooses to return multiple masks to match different levels of plausible results. Unlike SAM's solution, we use semantic-prompt to clarify the targets. As shown in Figure 4, the captions below the images are the text prompts, and the masks in the images are the predictions of SegVol, which show that semantic-prompt can effectively disambiguate the spatial-prompt.

**Identifying the spatial-prompt segmentation.** Furthermore, we study the capability of SegVol to identify the semantic category of the spatial-prompt results. Figure 5 reveals that SegVol can give accurate semantic categories based on the spatial-prompt results. In the top left image in Figure 5, the spatial-prompt on the liver results in a 0.997 prediction score for the liver. The top right image in the sub-figure shows if the spatial-prompt is the point on the liver tumor, SegVol will output a 0.619 prediction score for the tumor category and a 0.339 prediction score for the liver based on the spatial relationship of liver tumor and liver. We implement this identification experiment by decoding the semantic prompts from a category set. The softmax function is applied to the decoding results to get the prediction probabilities of different categories. The probabilities on the initial predicted mask, driven by the spatial-prompt, are used to calculate the final classification result.

## 4 Discussion

**Scalability.** The scaling law of foundation models has been verified in multiple CV and NLP tasks. Since SegVol uses a transformer-based architecture and self-supervised pre-training algorithm, it has strong data and architecture scalability. In this work, we achieve the success of scaling law in 3D medical segmentation by the design of universal prompts and pseudo masks for joint learning on datasets with inconsistent annotations. The ablation study of scaling up training data shows that 1) the performance improves significantly with more training data in the 3D segmentation task, 2) SegVol has not yet reached its ceiling if more training data is provided. We believe the performance of SegVol can be continuously improved when more data and computational resources are used.

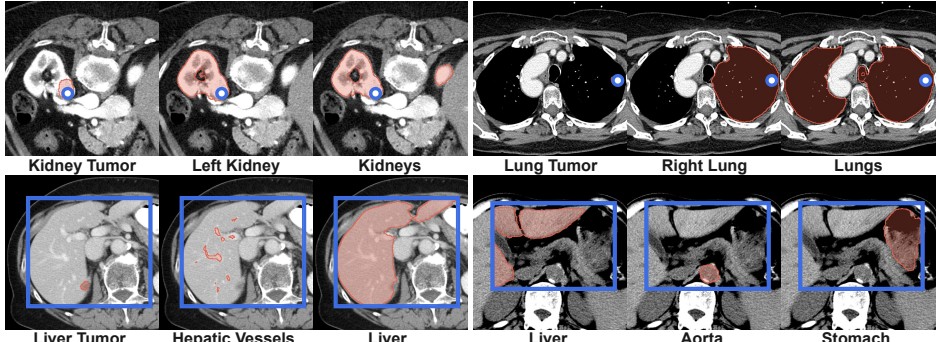

Figure 4: The four cases demonstrate that semantic-prompt can clarify the ambiguity of spatial-prompt and avoid multi-plausible outputs. Each image shows the segmentation result of SegVol using the spatial-prompt, i.e. point or bounding box, and semantic-prompt, i.e. the caption below the image.

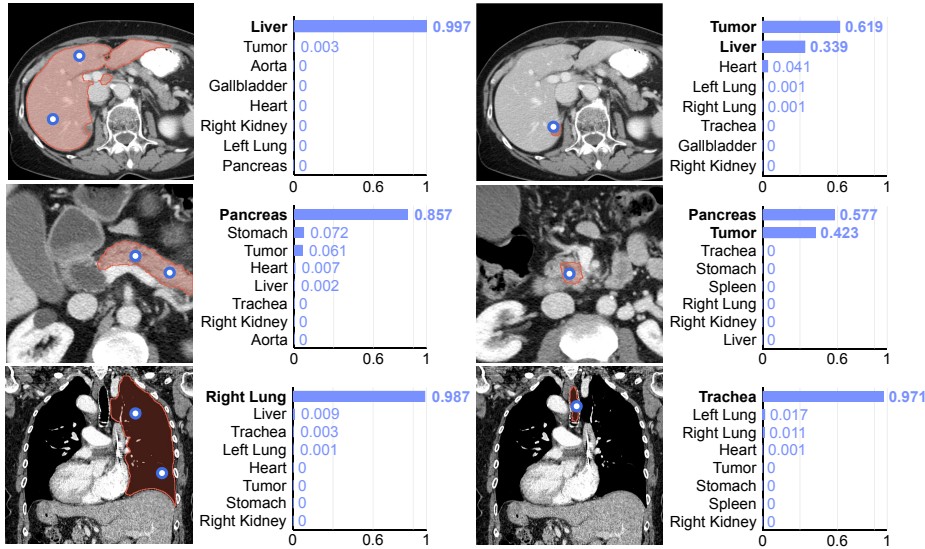

Figure 5: We identify the semantic categories of the spatial-prompt segmentation results. Each image shows the spatial-prompt and the mask prediction. The bar charts rank the top 8 semantic categories with the highest classification probabilities. The results show that SegVol is capable of identifying the anatomical category of the segmentation mask using spatial prompts.

**Generalizability to unseen modality.** Although we develop SegVol on Computed Tomography (CT) data due to its advantages of easy acquisition, wide usage, and high resolution, we find that SegVol can generalize to other medical image modality, like MRI. Namely, the SegVol model trained only on CT data can be used to segment MRI with semantic and spatial prompts. This emerging ability demonstrates that our foundation model understands the anatomical structure of human body. We provide detailed experiments and analysis of this generalizability in Section C. The impressive generalizability makes SegVol a versatile tool in medical image analysis. We leave the joint training of SegVol on multi-modality data as the future work.

**Limitations.** Although SegVol shows remarkable semantic-prompt segmentation performance, there still remains gap between it and the referring volumetric segmentation. A promising solution is to construct the referring segmentation data with diverse semantic and spatial prompts, and then train SegVol on it. We leave it as the future work. More discussions can be found in Section E.

**Broader impact.** We contribute a foundation model for universal and interactive volumetric medical image segmentation, which can benefit numerous clinical study and applications. As a foundational research work, we do not see any obvious negative societal impact of the proposed method and model.

## 5  Conclusion

In this paper, we propose SegVol, a universal and interactive volumetric medical image segmentation model, supporting both spatial-prompt and semantic-prompt segmentation of more than 200 anatomical categories. We construct a large-scale dataset, which consists of 90K unlabeled CTs and 25 open-source medical datasets, to train the foundation model. We design the zoom-out-zoom-in mechanism to facilitate efficient and precise inference in the region of interest. Extensive experiments on 22 segmentation tasks demonstrate the outstanding performance of our method. Detailed ablation studies are also carried out to prove the effectiveness of the zoom-out-zoom-in mechanism, dataset scale, and multi-prompt combination strategy. As a foundation model, we believe that SegVol will advance the volumetric medical segmentation and benefit numerous downstream tasks.

**Acknowledgements.** This work is funded by NSFC-62306046.

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

# A    Dataset Details and Availability

In this work, we collect 25 open-source datasets for supervised fine-tuning SegVol and some external open-source datasets specifically for comparative experiments. The detailed information on anatomical categories and dataset scales of these open-source datasets is shown in Table 4. The availability of these datasets is demonstrated in Table 5. Additionally, to avoid privacy concerns, we collect 90K unlabeled CTs from publicly accessible professional medical websites: https://radiopaedia.org/.

The collected segmentation datasets include major regions of the human body, i.e., the head, neck, thorax, abdomen, and pelvis, comprising over 200 categories of organs and tissues, and 28 lesion tasks from different benchmarks. The detailed categories information are shown in Figure 6 and some representative samples are shown in Figure 7.

Table 4: Information of datasets involved in supervised fine-tuning and experiments.

| Dataset | Anatomical Targets | Category Number | Trainset Volumes |
|---|---|---|---|
| 3D-IRCADB[55] | Liver and liver tumor | 47 | 20 |
| AbdomenCT-1k[45] | Liver, kidney, spleen, and pancreas | 4 | 1000 |
| AMOS22[44] | Abdominal organs | 15 | 240 |
| BTCV[52] | Abdominal organs | 13 | 30 |
| CHAOS[40, 41, 42] | Abdominal organs | 1 | 20 |
| CT-ORG[33, 34, 24, 35] | Brain, lung, bones, liver, kidney, and bladder | 6 | 140 |
| FLARE22[56, 57] | Thoracic and abdominal organs | 13 | 50 |
| HaN-Seg[43] | Organs of the head and neck | 30 | 42 |
| KiPA22[47, 48, 49, 50] | Kidney, renal tumor, artery, and vein | 4 | 70 |
| KiTS19[51] | Kidney and kidney tumor | 2 | 210 |
| KiTS23[46] | Kidney, kidney tumor, and kidney cyst | 3 | 489 |
| LUNA16[36] | Left lung, right lung, and trachea | 3 | 888 |
| MSD-Colon[56] | Colon tumor | 1 | 126 |
| MSD-HepaticVessel[56] | Hepatic vessel and liver tumor | 2 | 303 |
| MSD-Liver[56] | Liver and liver tumor | 2 | 131 |
| MSD-lung[56] | Lung tumor | 1 | 63 |
| MSD-pancreas[56] | Pancreas and pancreas tumor | 2 | 281 |
| MSD-spleen[56] | Spleen | 1 | 41 |
| Pancreas-CT[53, 54, 35] | Pancreas | 1 | 82 |
| QUBIQ[63] | Kidney, pancreas, and pancreas lesion | 3 | 82 |
| SegTHOR[75] | Heart, trachea, aorta, and esophagus | 4 | 40 |
| SLIVER07[62] | Liver | 1 | 20 |
| TotalSegmentator[58] | Organs of the whole body | 104 | 1203 |
| ULS23(novel annotated set)[74] | Various lesions | - | 1618 |
| VerSe19[59, 60, 61] | Vertebrae | 28 | 80 |
| VerSe20[59, 60, 61] | vertebrae | 28 | 61 |
| WORD[64] | Thoracic and abdominal organs | 16 | 100 |

# B    Training Algorithm

Due to the complexity of the training steps, which include decoder reuse, the combination of different datasets, and cooperative training of ground-truth and pseudo labels, we abstract the core training code as Algorithm 1 and Figure 8 to clarify the training process of SegVol. As shown in Figure 8, each case (training sample) consists of an Image $x$, a Ground Truth(GT) Mask Set $Y$, and a Pseudo Mask Set $Z$. The training loss of each sample consists of the ground-truth loss and the pseudo loss. The ground-truth loss is computed by inputting the image, the ground-truth mask (label), and the sampled prompt into the model, while the pseudo loss is computed by inputting the image, the pseudo

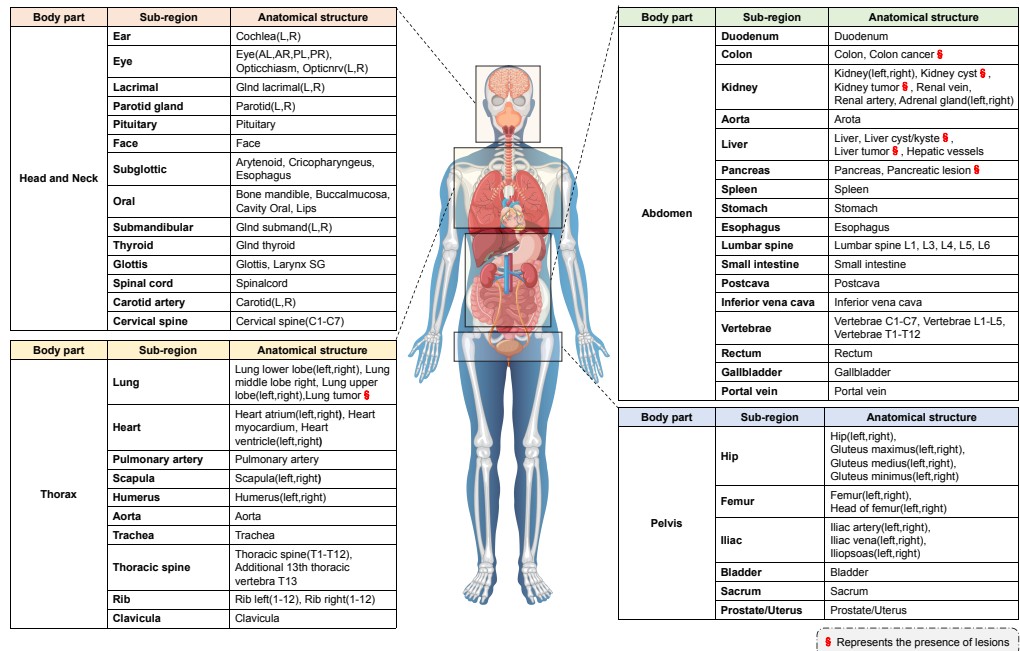

Figure 6: Overview of the collected datasets for supervised fine-tuning. The joint dataset comprises 47 important regions, with each region containing one or multiple significant anatomical structures within that spatial area. Image of the human body by brgfx on Freepik[76].

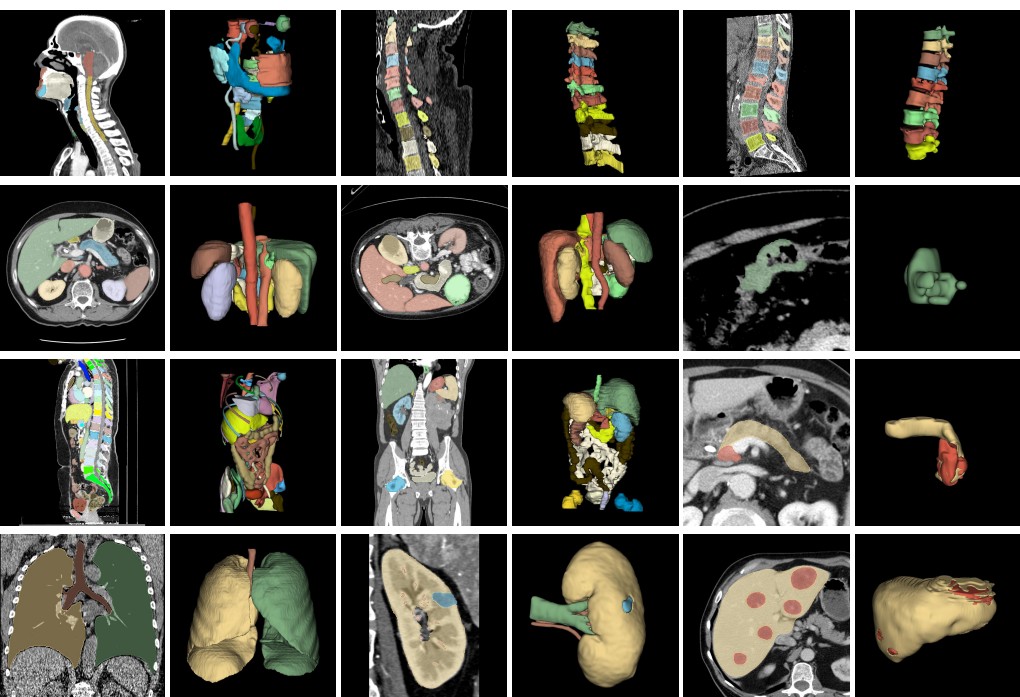

Figure 7: The joint dataset encompasses various anatomical structures in major regions of the human body. Several volume examples are demonstrated as 2D slices and 3D shapes in the images respectively.

Table 5: Availability of datasets involved in supervised fine-tuning and experiments.

| Dataset | Link |
|---------|------|
| 3D-IRCADB[55] | https://www.kaggle.com/datasets/nguyenhoainam27/3dircadb |
| AbdomenCT-1k[45] | https://github.com/JunMa11/AbdomenCT-1K |
| AMOS22[44] | https://amos22.grand-challenge.org/ |
| BTCV[52] | https://www.synapse.org/#!Synapse:syn3193805/wiki/217752 |
| CHAOS[40, 41, 42] | https://chaos.grand-challenge.org/ |
| CT-ORG[33, 34, 24, 35] | https://wiki.cancerimagingarchive.net/pages/viewpage.action?pageId=61080890 |
| FLARE22[56, 57] | https://flare22.grand-challenge.org/ |
| HaN-Seg[43] | https://han-seg2023.grand-challenge.org/ |
| KiPA22[47, 48, 49, 50] | https://kipa22.grand-challenge.org/ |
| KiTS19[51] | https://kits19.grand-challenge.org/ |
| KiTS23[46] | https://kits-challenge.org/kits23/ |
| LUNA16[36] | https://luna16.grand-challenge.org/Data/ |
| MSD-Colon[56] | http://medicaldecathlon.com/ |
| MSD-HepaticVessel[56] | http://medicaldecathlon.com/ |
| MSD-Liver[56] | http://medicaldecathlon.com/ |
| MSD-lung[56] | http://medicaldecathlon.com/ |
| MSD-pancreas[56] | http://medicaldecathlon.com/ |
| MSD-spleen[56] | http://medicaldecathlon.com/ |
| Pancreas-CT[53, 54, 35] | https://wiki.cancerimagingarchive.net/display/public/pancreas-ct |
| QUBIQ[63] | https://qubiq.grand-challenge.org/ |
| SegTHOR[75] | https://competitions.codalab.org/competitions/21145 |
| SLIVER07[62] | https://sliver07.grand-challenge.org/ |
| TotalSegmentator[58] | https://github.com/wasserth/TotalSegmentator |
| ULS23[74] | https://uls23.grand-challenge.org/ |
| VerSe19[59, 60, 61] | https://osf.io/nqjyw/ |
| VerSe20[59, 60, 61] | https://osf.io/t98fz/ |
| WORD[64] | https://paperswithcode.com/dataset/word |

Table 6: Complexity comparison of popular methods.

| Method | Total Parameters | Avg. MACs Per Case↓ | Avg. Time Per Case(s)↓ | Avg. Dice Score↑ |
|--------|------------------|---------------------|------------------------|------------------|
| SAM[28] | 94M | 1.3e+13 | 2.1764 | 0.5271 |
| MedSAM[29] | 94M | 1.3e+13 | 2.1886 | 0.5371 |
| SAM-MED2D[38] | 271M | 2.3e+12 | 3.5547 | 0.6668 |
| SAM-MED3D[39] | 101M | **1.0e+11** | **0.1768** | 0.6200 |
| SegVol | 181M | 6.7e+11 | 0.3283 | **0.8593** |

label, and the pre-designed prompt into the model. Finally, the model is optimized by minimizing the weighted sum of the two losses.

Besides, we add a reinforcement branch for semantic-prompt in the mask decoder. We further compute a similarity matrix between the up-scaled embedding from the transposed convolution output and the text embedding. The element-wise multiplication of the similarity matrix with the mask prediction is applied before interpolation, after which the model outputs the masks.

---
**Algorithm 1** SegVol training loop
---
**Input:** SegVol model, training image $x$, ground truth mask set $Y_x = \{y_i\}_{i=1}^n$, pseudo mask set
$\quad Z_x = \{z_i\}_{i=1}^m$
**Output:** SegVol model parameters
 1: $n \Leftarrow 6$ **# Number of combinations of 3 prompt types: text, point, and bbox.**
 2: $\alpha \Leftarrow 0.1$ **# Pseudo loss weight.**
 3: **# Loop for each category of this case.**
 4: **for** $i \Leftarrow 1$ **to** $n$ **do**
 5: $\quad f_{\text{img}} \Leftarrow \text{model.ImageEncoder}(x)$
 6: $\quad pt_{\text{spatial}}, pt_{\text{semantic}}, \Leftarrow \text{prompt\_generate}(y_i)$
 7: $\quad l_{\text{gt}} \Leftarrow 0$
 8: $\quad$ **# Loop for possible prompt combination types of ground truth mask.**
 9: $\quad$ **for** $p \Leftarrow 1$ **to** $n$ **do**
10: $\quad\quad$ **# Choose prompt combination type.**
11: $\quad\quad pt'_{\text{spatial}}, pt'_{\text{semantic}} \Leftarrow \text{PromptStrategy}(pt_{\text{spatial}}, pt_{\text{semantic}})$
12: $\quad\quad f_{\text{text}} \Leftarrow \text{model.TextEncoder}(pt'_{\text{semantic}})$
13: $\quad\quad f_{\text{prompt}} \Leftarrow \text{model.PromptEncoder}(pt'_{\text{spatial}}, f_{\text{text}})$
14: $\quad\quad pred_{\text{gt}} \Leftarrow \text{model.Decoder}(f_{\text{img}}, f_{\text{prompt}}, f_{\text{text}})$
15: $\quad\quad l_{\text{gt}} \Leftarrow l_{\text{gt}} + \text{DiceLoss}(pred_{\text{gt}}, y_i) + \text{BCELoss}(pred_{\text{gt}}, y_i)$
16: $\quad$ **end for**
17: $\quad l_{\text{pseudo}} \Leftarrow 0$
18: $\quad$ **# Loop for several pseudo masks.**
19: $\quad$ **for** $p \Leftarrow 1$ **to** $n$ **do**
20: $\quad\quad$ **# Random select a pseudo mask of this case for training.**
21: $\quad\quad z_p \Leftarrow \text{RandomSelect}(Z_x, [1, m])$
22: $\quad\quad pt_{\text{spatial}} \Leftarrow \text{prompt\_generate}(z_p)$
23: $\quad\quad f_{\text{prompt}} \Leftarrow \text{model.PromptEncoder}(pt_{\text{spatial}})$
24: $\quad\quad pred_{\text{pseudo}} \Leftarrow \text{model.Decoder}(f_{\text{img}}, f_{\text{prompt}})$
25: $\quad\quad l_{\text{pseudo}} \Leftarrow l_{\text{pseudo}} + \text{DiceLoss}(pred_{\text{pseudo}}, z_p) + \text{BCELoss}(pred_{\text{pseudo}}, z_p)$
26: $\quad$ **end for**
27: $\quad l \Leftarrow l_{\text{gt}} + \alpha \times l_{\text{pseudo}}$
28: $\quad \text{update}(\text{model}, l)$
29: **end for**
30: **return** model
---

## C   Additional Experimental Analysis

**Comparative experiments to compare with task-specific segmentation models.**   Task-specific segmentation models mainly fall into two architectures, CNN-based models and Transformer-based models. We conduct comparative experiments with representative CNN-based models i.e. 3DUX-Net[23] and nnU-Net[22], and representative Transformer-based models i.e. SwinUNETR[20]. We conduct additional comparative experiments on the split 20% test set of the datasets. 10 segmentation tasks are selected from BTCV[52] and MSD-spleen[56] datasets, which focus on organ segmentation, and from MSD-lung, MSD-colon, and MSD-liver datasets, which focus on lesion segmentation. We train task-specific segmentation models on each dataset individually for each method.

The quantitative experimental results are summarized in Figure 9. Generally speaking, SegVol, jointly trained on 25 datasets, outperforms traditional task-specific segmentation models trained on a single dataset. Compared to these strong baselines, SegVol exhibits a narrower distribution of Dice scores across the eight tasks, indicating its robustness and good generalization ability. This mainly owes to the massive knowledge learned from diverse samples of the same categories but different datasets. SegVol depicts excellent performance on lesion tasks which are more challenging in semantic understanding and spatial locating. We present a detailed comparison to nnU-Net[22] on lesion tasks. As shown in Table 7, the average Dice score of SegVol is 14.76% higher than that of nnU-Net for lesion tasks. We visualize the prediction results of the two methods in Figure 10, which intuitively show that SegVol performs more precise segmentation of the tumors than nnU-Net. The detailed scores and visualization results are presented in Table 9 and Figure 11 12, and 13.

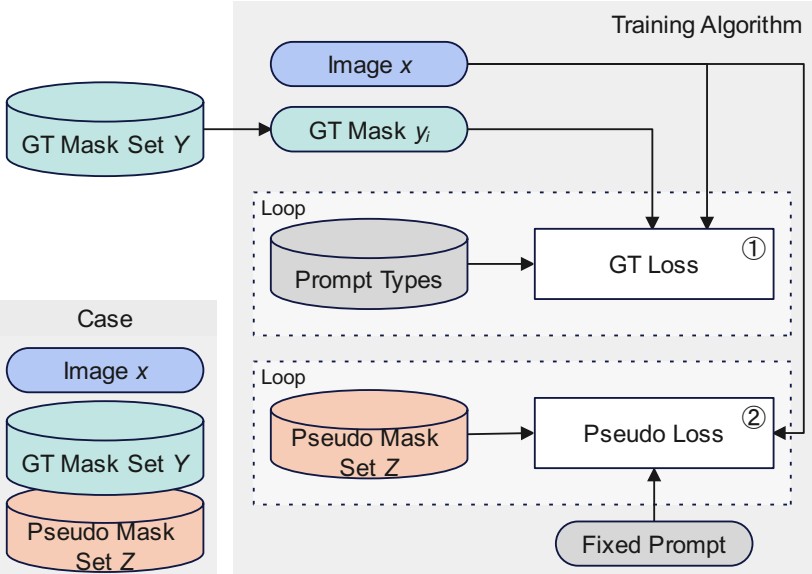

Figure 8: The demonstration of the training algorithm. Specifically, each case (training sample) consists of an Image $x$, a Ground Truth(GT) Mask Set $Y$, and a Pseudo Mask Set $Z$. The training loss of each sample consists of the ground-truth loss and the pseudo loss. The ground-truth loss is computed by inputting the image, the ground-truth mask (label), and the sampled prompt into the model, while the pseudo loss is computed by inputting the image, the pseudo label, and the fixed prompt into the model. Finally, the model is updated by minimizing the weighted sum of the two losses.

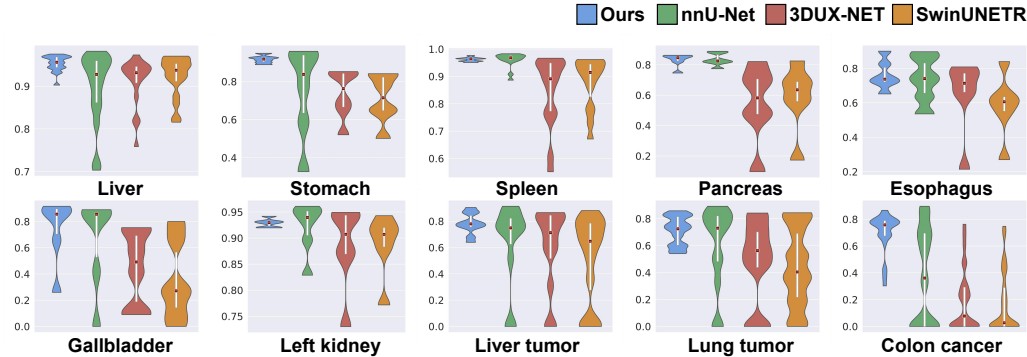

Figure 9: Violin plots for comparing experiment results of SegVol and task-specific methods. The vertical axis is the Dice score.

We analyze that there are mainly three factors that make SegVol more powerful than traditional task-specific models: 1) Massive generative pre-training on unlabeled data endows SegVol with a complete understanding of the volumetric structures and the discriminative feature representations, which is much superior to learning from a small number of samples. 2) Learning from joint datasets with semantic-prompt makes SegVol generalize better to unseen data and categories. For instance, SegVol can learn from both the 'left kidney' and 'kidney' categories based on their semantic correlation, while traditional task-specific models treat the two categories independently. 3) SegVol can be prompted with (spatial) points/bboxes, which provide a precise spatial reference, and (semantic) texts, which disambiguate the overlap of multiple categories in the same space. In contrast, traditional methods are not able to understand semantics. This ability enables SegVol to perform better than traditional methods in challenging tasks, e.g., segmenting lesions.

Table 7: The comparison of the average Dice score of SegVol and nnU-Net[22] across 3 lesion segmentation tasks.

| Method | Lung Tumor | Colon Cancer | Liver Tumor |
|---|---|---|---|
| nnU-Net[22] | 0.5963 | 0.3769 | 0.3606 |
| **OURS** | **0.7122** | **0.6965** | **0.7825** |

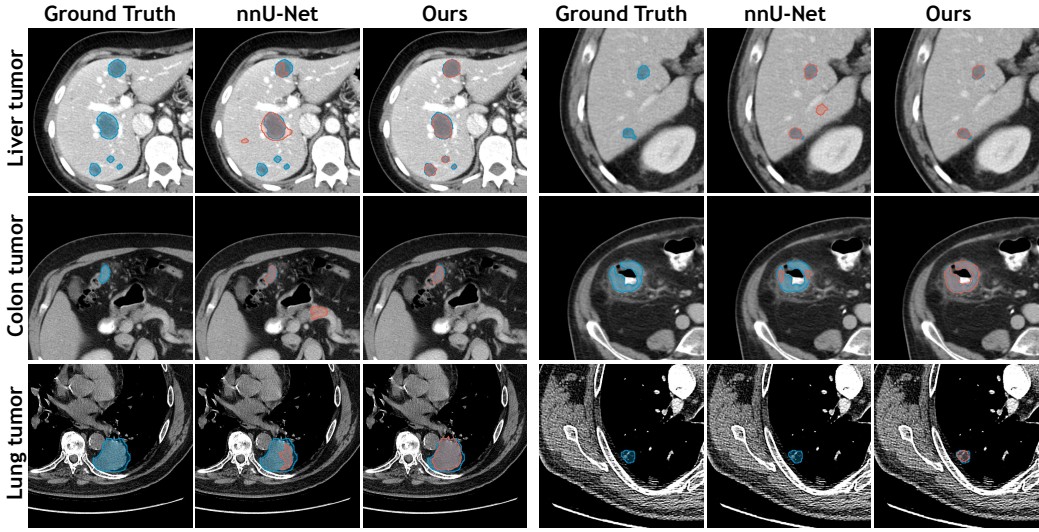

Figure 10: Visualization results of SegVol and nnU-Net across 3 lesion segmentation tasks.

**Supplement results for comparative experiments on SAM-like interactive segmentation methods.**
In this work, we compare SegVol with 5 SAM-like interactive segmentation methods on AMOS22[44], ULS23[74], and SegTHOR[75] datasets. The detailed records of Dice score are demonstrated in Table 10. The visualization results are shown in Figure 14, Figure 15, Figure 16. In this experiment, SegVol is driven by 'bbox+text' prompt. We demonstrate the consistency results among different prompt settings of SegVol in Figure 17, which is also conducted on AMOS22[44] and ULS23[74]. Relatively poor text prompt results in ULS23 are due to the unclear category of the dataset.

We also compare the Total Parameters, the average Multiply-Accumulates(MACs), and the average Time required to process a case of the different SAM-like methods, as shown in Table6. The comparison indicates that our method takes less computational cost while achieving much better performance. Note that when calculating MACs Per Case and Time Per Case, the slice-by-slice calculation of the 2D method and the scanning process of the 3D method are accumulated respectively. SAM-MED3D[39] only processes volume with a size of $128 \times 128 \times 128$. The experiments are implemented on the validation set of AMOS22[44], and the setting is the same as that in Sec. 3.2

**Supplement results for ablation studies on zoom-out-zoom-in mechanism.** We conduct ablation study on zoom-out-zoom-in mechanism on the split 20% test data of AMOS22[44] dataset. As shown in Table 11, the zoom-out-zoom-in mechanism achieves higher Dice scores compared to the

Table 8: Few-shot fine-tuning experiment on FLARE22[56, 57] and MSD-spleen[56]. **SegVol\*** represents the model fine-tuned on all datasets.

| Avg. Dice Score | 100 epochs | 200 epochs | 300 epochs | 400 epochs | 500 epochs | SegVol* |
|---|---|---|---|---|---|---|
| FLARE22[56, 57] | 0.0463 | 0.4028 | 0.4926 | 0.5617 | 0.5567 | 0.8822 |
| MSD-spleen[56] | 0.7566 | 0.7866 | 0.9433 | 0.9454 | 0.9471 | 0.9597 |

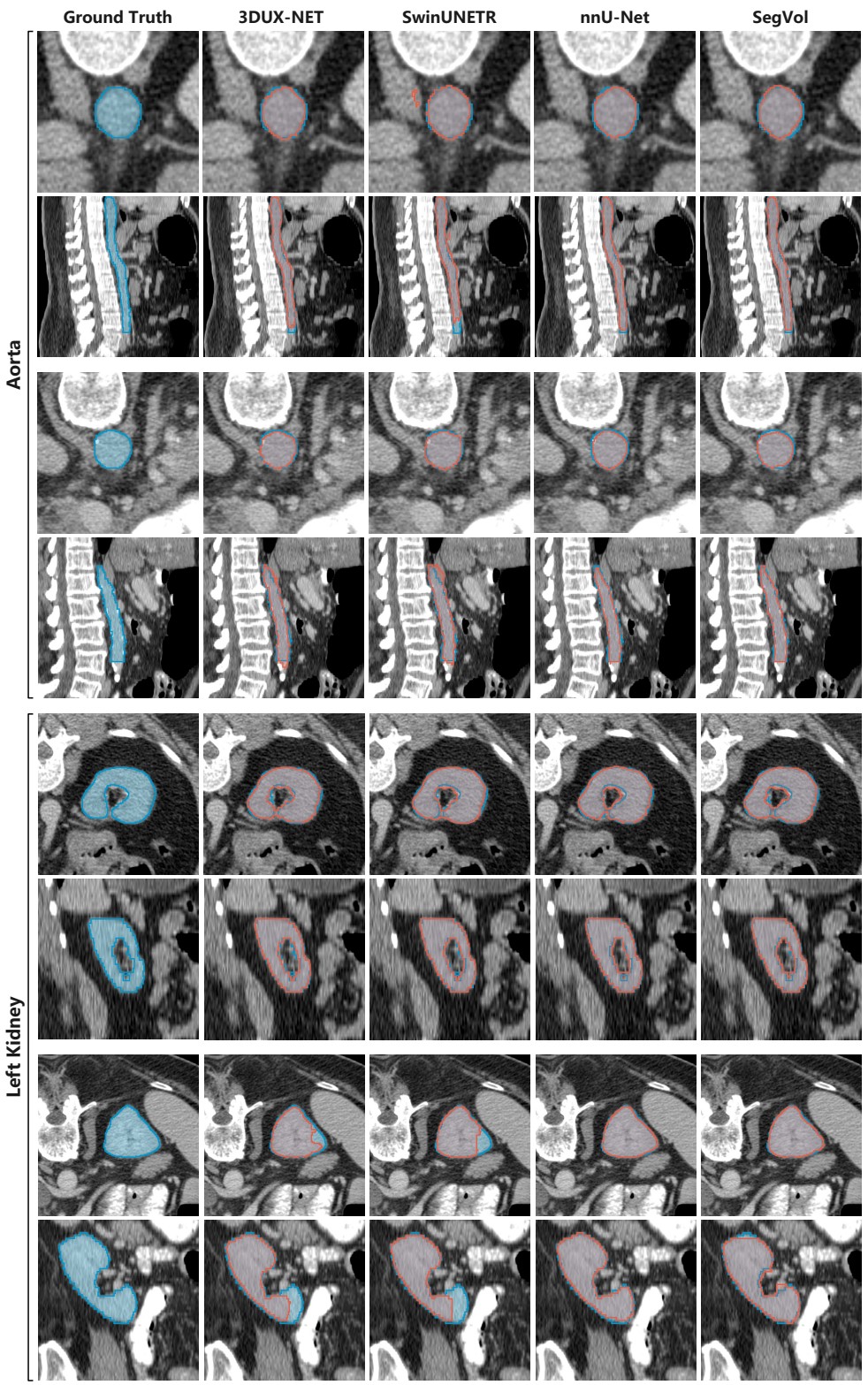

Figure 11: Visualized aorta and left kidney prediction results of 3DUX-NET[23], SwinUNETR[20], nnU-Net[22] and SegVol on 4 cases from the split test set. For the integrality of aorta and left kidney structure modeling, SegVol significantly outperforms 3DUX-NET and SwinUNETR and is comparable to nnU-Net.

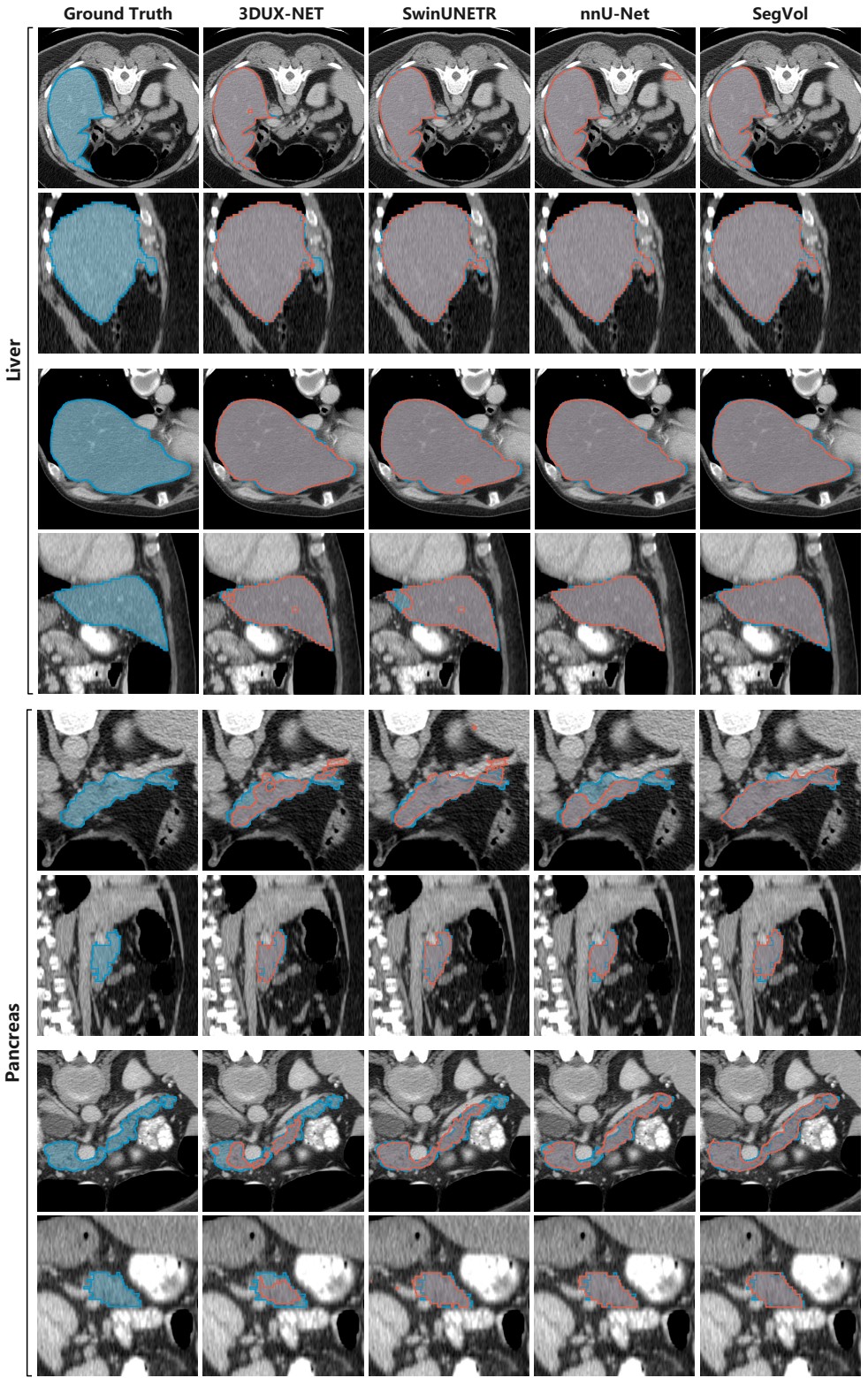

Figure 12: Visualized liver and pancreas prediction results of 3DUX-NET[23], SwinUNETR[20], nnU-Net[22] and SegVol on 4 cases from the split test set. For the modeling of pancreas, SegVol is significantly superior to other baseline methods.

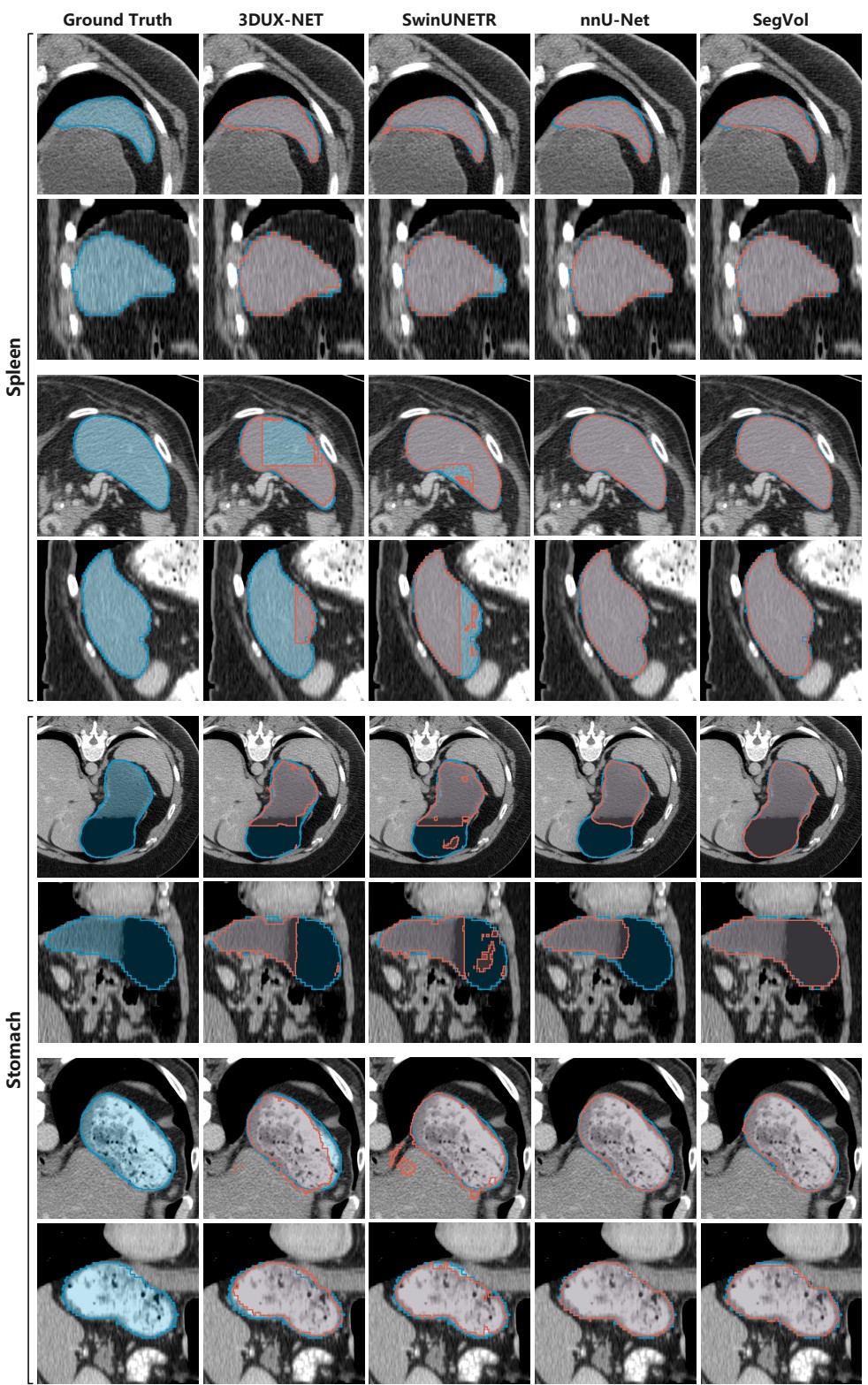

Figure 13: Visualized spleen and stomach prediction results of 3DUX-NET[23], SwinUNETR[20], nnU-Net[22] and SegVol on 4 cases from the split test set. For the consistency and stability of stomach modeling, SegVol is significantly better than other methods.

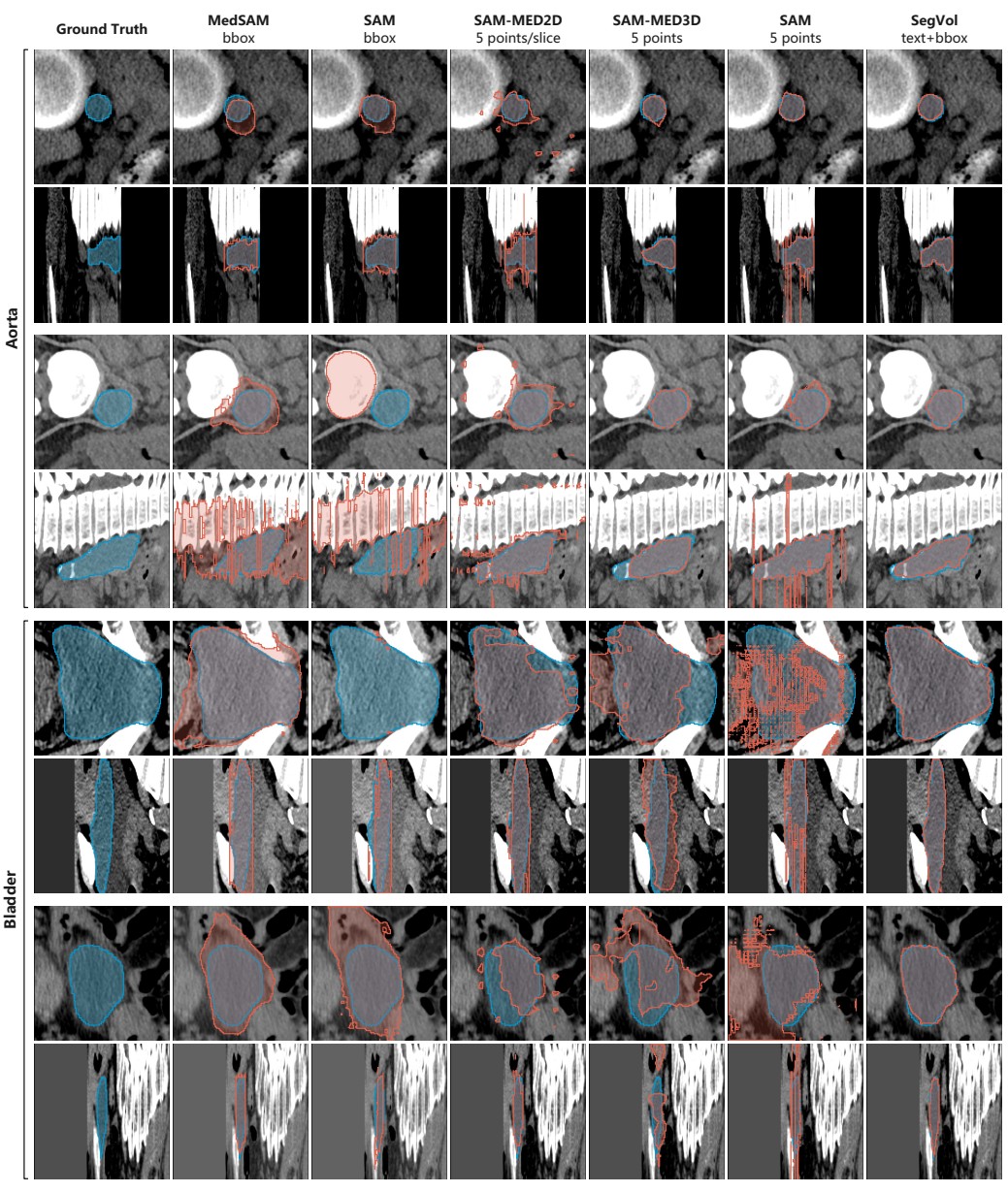

Figure 14: Visualized aorta and bladder prediction results of MedSAM[29], SAM(bbox)[28], SAM-MED2D[38], SAM-MED3D[39], SAM(points)[28] and SegVol on 4 cases from split test data.

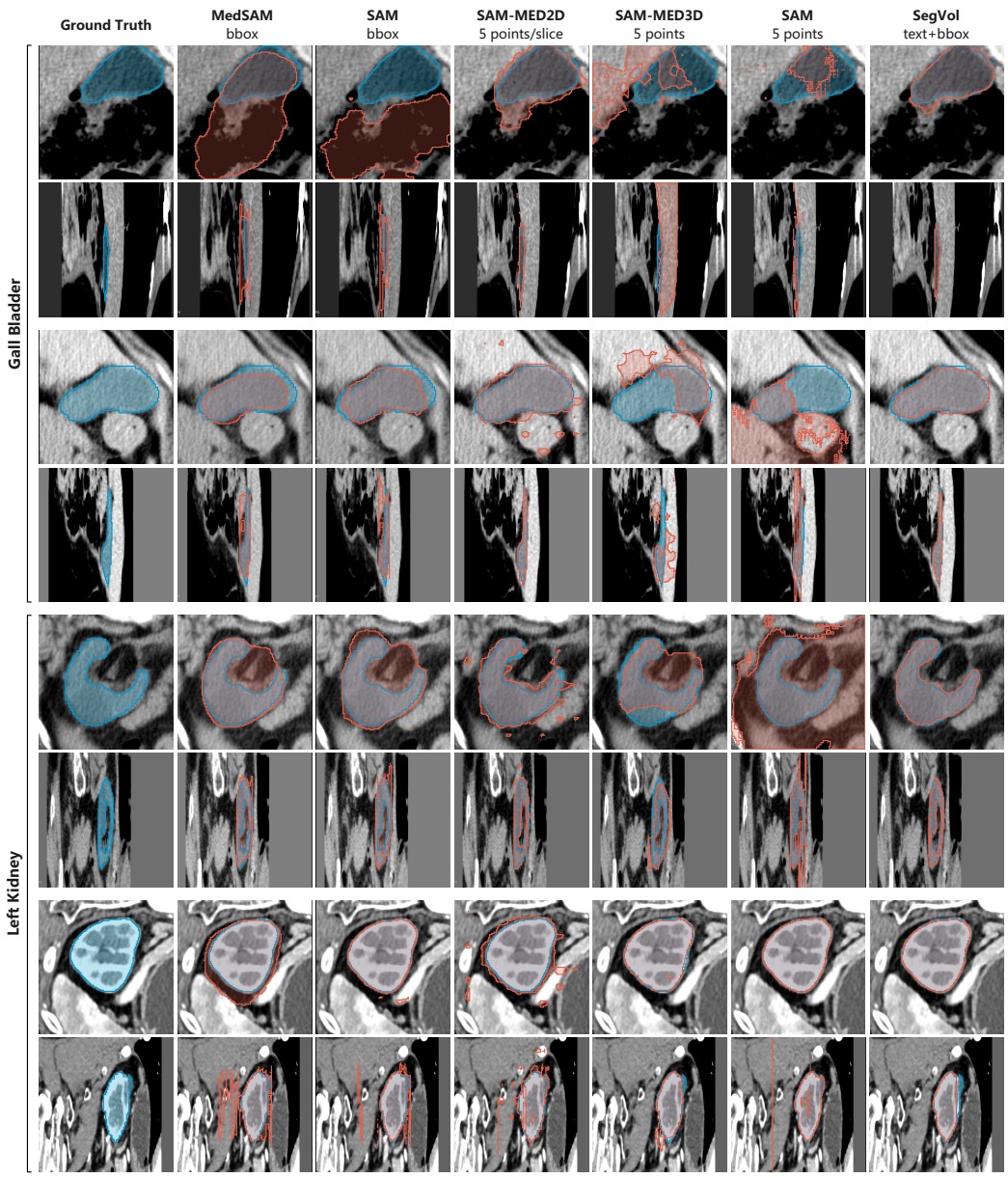

Figure 15: Visualized gall bladder and left kidney prediction results of MedSAM[29], SAM(bbox)[28], SAM-MED2D[38], SAM-MED3D[39], SAM(points)[28] and SegVol on 4 cases from split test data.

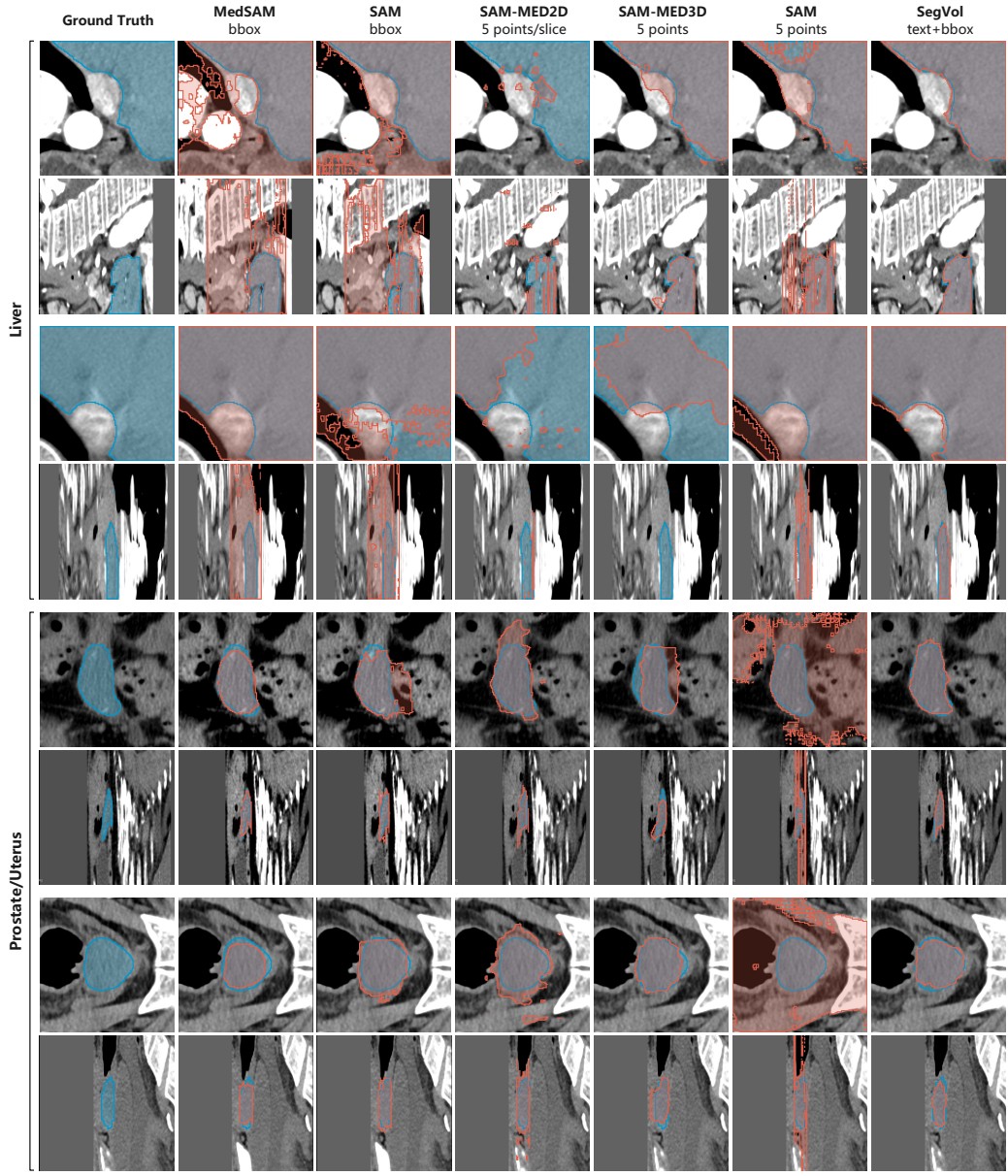

Figure 16: Visualized liver and prostate/uterus prediction results of MedSAM[29], SAM(bbox)[28], SAM-MED2D[38], SAM-MED3D[39], SAM(points)[28] and SegVol on 4 cases from split test data.

Table 9: Comparative experiment results of 3DUX-NET, SwinUNETR, nnU-Net, and SegVol on the test set of supervised fine-tuning datasets in terms of Dice score. Dice scores are displayed as 'Median values (First quartile, Third quartile)'.

| Category | 3DUX-NET[23] | SwinUNETR[20] | nnU-Net[22] | SegVol |
|---|---|---|---|---|
| Aorta | 0.9122 (0.8852, 0.9292) | 0.8870 (0.8619, 0.8964) | 0.9155 (0.8790, 0.9431) | 0.9179 (0.8850, 0.9256) |
| Colon cancer | 0.0773 (0.0000, 0.2931) | 0.0270 (0.0003, 0.2908) | 0.3610 (0.0000, 0.6961) | 0.7582 (0.6749, 0.7903) |
| Esophagus | 0.7136 (0.6617, 0.7718) | 0.6063 (0.5508, 0.6353) | 0.7407 (0.6563, 0.8313) | 0.7373 (0.7205, 0.8062) |
| Gallbladder | 0.4916 (0.1875, 0.6926) | 0.2714 (0.1421, 0.5671) | 0.8555 (0.5267, 0.8633) | 0.8560 (0.7036, 0.8968) |
| Inferior vena cava | 0.7673 (0.6740, 0.8465) | 0.7368 (0.6376, 0.8376) | 0.8138 (0.7580, 0.8487) | 0.8267 (0.8044, 0.8418) |
| Left adrenal gland | 0.5788 (0.3238, 0.6038) | 0.5658 (0.4380, 0.6147) | 0.7915 (0.6888, 0.8231) | 0.7643 (0.6525, 0.7880) |
| Left kidney | 0.9072 (0.8692, 0.9438) | 0.9070 (0.8829, 0.9203) | 0.9395 (0.9050, 0.9518) | 0.9296 (0.9228, 0.9321) |
| Liver | 0.9316 (0.9074, 0.9462) | 0.9374 (0.9110, 0.9531) | 0.9276 (0.8614, 0.9597) | 0.9560 (0.9437, 0.9685) |
| Liver tumor | 0.7131 (0.5159, 0.8457) | 0.6479 (0.2756, 0.7853) | 0.7495 (0.6243, 0.8228) | 0.7801 (0.7558, 0.8440) |
| Lung tumor | 0.5628 (0.4375, 0.7021) | 0.4043 (0.2159, 0.6910) | 0.7294 (0.4814, 0.8210) | 0.7250 (0.6026, 0.8154) |
| Pancreas | 0.5820 (0.4748, 0.7069) | 0.6352 (0.5586, 0.6894) | 0.8248 (0.8169, 0.8665) | 0.8464 (0.8248, 0.8578) |
| Portal vein & splenic vein | 0.7207 (0.6211, 0.7588) | 0.6656 (0.5888, 0.6982) | 0.7964 (0.7524, 0.8582) | 0.7188 (0.7128, 0.7569) |
| Right adrenal gland | 0.5785 (0.5099, 0.6302) | 0.5026 (0.2730, 0.5963) | 0.7137 (0.7067, 0.7326) | 0.6579 (0.6372, 0.7008) |
| Right kidney | 0.9177 (0.8877, 0.9417) | 0.9065 (0.9011, 0.9289) | 0.9432 (0.9207, 0.9504) | 0.9227 (0.9157, 0.9295) |
| Spleen | 0.8913 (0.7726, 0.9492) | 0.9147 (0.8255, 0.9456) | 0.9681 (0.9596, 0.9766) | 0.9642 (0.9558, 0.9664) |
| Stomach | 0.7627 (0.6655, 0.8424) | 0.7147 (0.6470, 0.8231) | 0.8374 (0.6339, 0.9391) | 0.9177 (0.9035, 0.9260) |

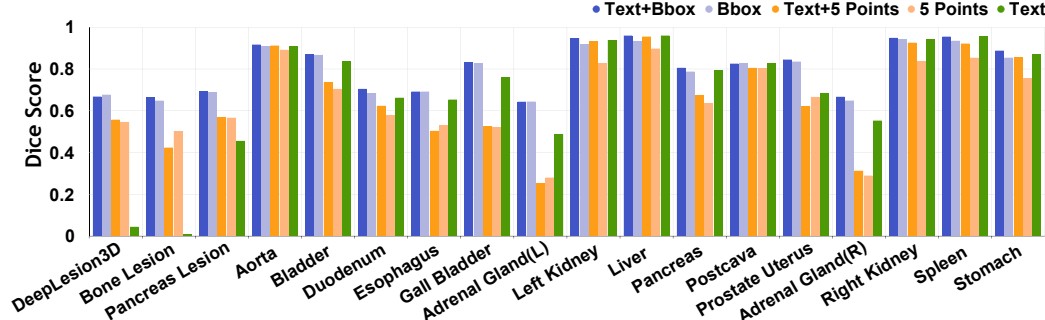

Figure 17: The bar chart illustrates the consistency of SegVol's performance across different prompt types on the validation set of AMOS22[44] and ULS23[74].

resize and sliding window strategies in 15 organ categories. In the aspect of inference efficiency, the zoom-out-zoom-in mechanism is also very competitive and quite close to the simple resize method.

**Generalization performance of SegVol on MRI.** We discuss the generalization performance of SegVol on an external MRI dataset. We collect 60 MRI scans annotated with 4 key organ categories from CHAOS[40, 41, 42] dataset and evaluate the generalization ability to unseen modality of SegVol. It achieves median Dice scores of 85.70%, 80.09%, 80.04%, and 81.46% for liver, spleen, left kidney, and right kidney, respectively. This generalization result demonstrates the robustness of SegVol in the face of completely unseen modality data. The detailed scores and visualization results are presented in Table 12 and Figure 18.

**Few-shot fine-tuning experiment on small datasets.** To evaluate the few-shot learning ability of our model, we conduct the few-shot fine-tuning experiment on small datasets, FLARE22[56, 57] (40 training cases) and MSD-spleen[56] (32 training cases). Table 8 demonstrates that 1) fine-tuning SegVol on dozens of samples works well on easy datasets such as MSD-spleen, in which the few-shot learning performance is close to the joint fine-tuning on all datasets; 2) for challenging datasets such as FLARE22, fine-tuning on all datasets can achieve much better performance.

Table 10: Comparative experiment results of SAM(Point), SAM(Bbox), SAM-MED2D, SAM-MED3D, MedSAM, and SegVol in terms of Dice score. Dice scores are displayed as 'Median values (First quartile, Third quartile)'. The three columns in the table are, in order, the AMOS22[44], ULS23[74], and SegTHOR[75] datasets.

| Category | SAM(Point)[28] | SAM(Bbox)[28] | SAM-MED2D[38] | SAM-MED3D[39] | MedSam[29] | SegVol |
|---|---|---|---|---|---|---|
| Aorta | 0.7267 (0.5213, 0.9350) | 0.4362 (0.3491, 0.5646) | 0.8704 (0.8260, 0.9141) | 0.8102 (0.6680, 0.8692) | 0.3387 (0.2778, 0.4478) | 0.9273 (0.9050, 0.9424) |
| Bladder | 0.4162 (0.2862, 0.5099) | 0.6281 (0.3093, 0.7565) | 0.8417 (0.7484, 0.9024) | 0.4338 (0.2445, 0.7198) | 0.6799 (0.4275, 0.7992) | 0.9120 (0.8338, 0.9446) |
| Duodenum | 0.1554 (0.1039, 0.2125) | 0.3192 (0.2559, 0.3886) | 0.5066 (0.4170, 0.5725) | 0.3820 (0.2427, 0.4981) | 0.3066 (0.2635, 0.3661) | 0.7402 (0.6594, 0.7909) |
| Esophagus | 0.2917 (0.1019, 0.6169) | 0.3541 (0.2167, 0.5540) | 0.5500 (0.4131, 0.6599) | 0.5174 (0.3678, 0.6792) | 0.3610 (0.2560, 0.5402) | 0.7460 (0.6376, 0.8115) |
| Gallbladder | 0.2831 (0.1756, 0.5198) | 0.6161 (0.4809, 0.7200) | 0.7999 (0.7097, 0.8725) | 0.5643 (0.3615, 0.7377) | 0.6609 (0.5446, 0.7245) | 0.8763 (0.8020, 0.9082) |
| Left adrenal gland | 0.0555 (0.0276, 0.2347) | 0.4222 (0.3417, 0.4995) | 0.5068 (0.3225, 0.6318) | 0.4584 (0.3104, 0.6267) | 0.3766 (0.3321, 0.4541) | 0.7295 (0.6519, 0.7916) |
| Left kidney | 0.8405 (0.6844, 0.9464) | 0.8274 (0.7733, 0.8631) | 0.9325 (0.8899, 0.9467) | 0.8723 (0.7705, 0.9286) | 0.7909 (0.7409, 0.8139) | 0.9489 (0.9389, 0.9585) |
| Liver | 0.7477 (0.6695, 0.8085) | 0.5124 (0.4467, 0.5801) | 0.6904 (0.5401, 0.8016) | 0.8801 (0.8204, 0.9321) | 0.6137 (0.5783, 0.6479) | 0.9641 (0.9547, 0.9701) |
| Pancreas | 0.2127 (0.1558, 0.3109) | 0.3392 (0.2572, 0.4243) | 0.5656 (0.5155, 0.6413) | 0.5391 (0.3304, 0.7333) | 0.3217 (0.2756, 0.4020) | 0.8295 (0.7734, 0.8711) |
| Postcava | 0.2042 (0.1402, 0.3478) | 0.5251 (0.4349, 0.5925) | 0.4436 (0.3029, 0.6463) | 0.6683 (0.5353, 0.7672) | 0.5211 (0.4598, 0.6180) | 0.8384 (0.7909, 0.8684) |
| Prostate uterus | 0.2344 (0.1655, 0.3081) | 0.6986 (0.5430, 0.7522) | 0.7518 (0.6567, 0.8261) | 0.6231 (0.5330, 0.7364) | 0.7739 (0.6685, 0.8271) | 0.8557 (0.8255, 0.8901) |
| Right adrenal gland | 0.0452 (0.0268, 0.1082) | 0.3642 (0.2766, 0.4491) | 0.1681 (0.0873, 0.3560) | 0.3708 (0.2454, 0.5182) | 0.3855 (0.3103, 0.4710) | 0.6994 (0.6138, 0.7661) |
| Right kidney | 0.8459 (0.5935, 0.9497) | 0.8215 (0.7528, 0.8577) | 0.9077 (0.8685, 0.9419) | 0.8632 (0.7755, 0.9258) | 0.7851 (0.7506, 0.8227) | 0.9505 (0.9426, 0.9585) |
| Spleen | 0.5936 (0.4686, 0.7846) | 0.6536 (0.5934, 0.7697) | 0.9267 (0.8821, 0.9483) | 0.8591 (0.7552, 0.9297) | 0.7038 (0.6609, 0.7766) | 0.9589 (0.9465, 0.9677) |
| Stomach | 0.4229 (0.3437, 0.5479) | 0.3883 (0.3051, 0.4713) | 0.5399 (0.4555, 0.6267) | 0.4576 (0.2540, 0.6447) | 0.4378 (0.3503, 0.5379) | 0.9123 (0.8677, 0.9369) |
| DeepLesion3D | 0.3686 (0.0855, 0.7680) | 0.7473 (0.6817, 0.8063) | 0.3258 (0.1325, 0.5707) | 0.2386 (0.1045, 0.4372) | 0.7680 (0.7103, 0.8160) | 0.7065 (0.6247, 0.7782) |
| BoneLesion | 0.4461 (0.2349, 0.6676) | 0.6671 (0.5854, 0.7443) | 0.1947 (0.0898, 0.3969) | 0.4447 (0.1481, 0.7026) | 0.6896 (0.6128, 0.7530) | 0.6920 (0.6097, 0.7702) |
| PancreasLesion | 0.0675 (0.0471, 0.1237) | 0.5579 (0.4911, 0.6111) | 0.5548 (0.4862, 0.6382) | 0.5526 (0.3287, 0.6786) | 0.6561 (0.5857, 0.7143) | 0.7265 (0.6722, 0.7776) |
| Aorta | 0.2744 (0.2289, 0.3488) | 0.3894 (0.3356, 0.4325) | 0.8077 (0.7693, 0.8289) | 0.7703 (0.7090, 0.8133) | 0.3278 (0.2786, 0.3650) | 0.8439 (0.8246, 0.8673) |
| Esophagus | 0.0348 (0.0250, 0.0439) | 0.2046 (0.1416, 0.2455) | 0.3578 (0.2545, 0.4281) | 0.6394 (0.5477, 0.7267) | 0.2196 (0.1633, 0.2630) | 0.7201 (0.6512, 0.7598) |
| Heart | 0.6695 (0.6180, 0.7419) | 0.8876 (0.8652, 0.9073) | 0.6012 (0.4050, 0.6922) | 0.8326 (0.8059, 0.8536) | 0.8924 (0.8782, 0.9099) | 0.8172 (0.7664, 0.8609) |
| Trachea | 0.9147 (0.6688, 0.9545) | 0.1611 (0.1164, 0.2218) | 0.8306 (0.7501, 0.8642) | 0.8485 (0.8110, 0.8911) | 0.1261 (0.1012, 0.1648) | 0.8807 (0.8571, 0.9044) |

Table 11: Dice score and inference time results of ablation study on zoom-out-zoom-in mechanism.

| Category | Mechanism | Dice Score Avg.↑ | Time Per Case Avg.↓ |
|---|---|---|---|
| Arota | Resize | 0.6375 | 64 ms |
| | Zoom-out-zoom-in | 0.8853 | 341 ms |
| | Sliding window | 0.8715 | 3452 ms |
| Bladder | Resize | 0.3229 | 65 ms |
| | Zoom-out-zoom-in | 0.6530 | 135 ms |
| | Sliding window | 0.5146 | 3098 ms |
| Duodenum | Resize | 0.2821 | 64 ms |
| | Zoom-out-zoom-in | 0.6414 | 168 ms |
| | Sliding window | 0.5013 | 3442 ms |
| Esophagus | Resize | 0.1778 | 62 ms |
| | Zoom-out-zoom-in | 0.4530 | 180 ms |
| | Sliding window | 0.3372 | 3186 ms |
| Gall bladder | Resize | 0.3558 | 63 ms |
| | Zoom-out-zoom-in | 0.6830 | 140 ms |
| | Sliding window | 0.5758 | 3220 ms |
| Left adrenal gland | Resize | 0.0937 | 64 ms |
| | Zoom-out-zoom-in | 0.4542 | 134 ms |
| | Sliding window | 0.2080 | 2935 ms |
| Left kidney | Resize | 0.6472 | 64 ms |
| | Zoom-out-zoom-in | 0.9362 | 177 ms |
| | Sliding window | 0.9046 | 3440 ms |
| Liver | Resize | 0.8252 | 62 ms |
| | Zoom-out-zoom-in | 0.9593 | 287 ms |
| | Sliding window | 0.9516 | 3450 ms |
| Pancreas | Resize | 0.3444 | 65 ms |
| | Zoom-out-zoom-in | 0.7292 | 159 ms |
| | Sliding window | 0.6206 | 3447 ms |
| Postcava | Resize | 0.5882 | 63 ms |
| | Zoom-out-zoom-in | 0.7901 | 296 ms |
| | Sliding window | 0.7769 | 3453 ms |
| Prostate/uterus | Resize | 0.4072 | 65 ms |
| | Zoom-out-zoom-in | 0.6380 | 136 ms |
| | Sliding window | 0.5500 | 3034 ms |
| Right adrenal gland | Resize | 0.1660 | 64 ms |
| | Zoom-out-zoom-in | 0.5260 | 141 ms |
| | Sliding window | 0.3624 | 3442 ms |
| Right kidney | Resize | 0.6570 | 64 ms |
| | Zoom-out-zoom-in | 0.8909 | 175 ms |
| | Sliding window | 0.9017 | 3441 ms |
| Spleen | Resize | 0.6827 | 89 ms |
| | Zoom-out-zoom-in | 0.8942 | 199 ms |
| | Sliding window | 0.8941 | 3481 ms |
| Stomach | Resize | 0.5752 | 63 ms |
| | Zoom-out-zoom-in | 0.8136 | 181 ms |
| | Sliding window | 0.8238 | 3446 ms |

Table 12: Generalization experiment results of SegVol on the MRI set of CHAOS[40, 41, 42] dataset in term of Dice score. Dice scores are displayed as 'Median values (First quartile, Third quartile)'.

| Method | Liver | Spleen | Left Kidney | Right Kidney |
|---|---|---|---|---|
| SegVol(5 Points) | 0.8091 (0.7376, 0.8554) | 0.7496 (0.6990, 0.7872) | 0.7216 (0.6125, 0.7869) | 0.7174 (0.6052, 0.8090) |
| SegVol(Bbox) | 0.8570 (0.8319, 0.8819) | 0.8009 (0.7702, 0.8256) | 0.8004 (0.7265, 0.8452) | 0.8146 (0.7593, 0.8620) |

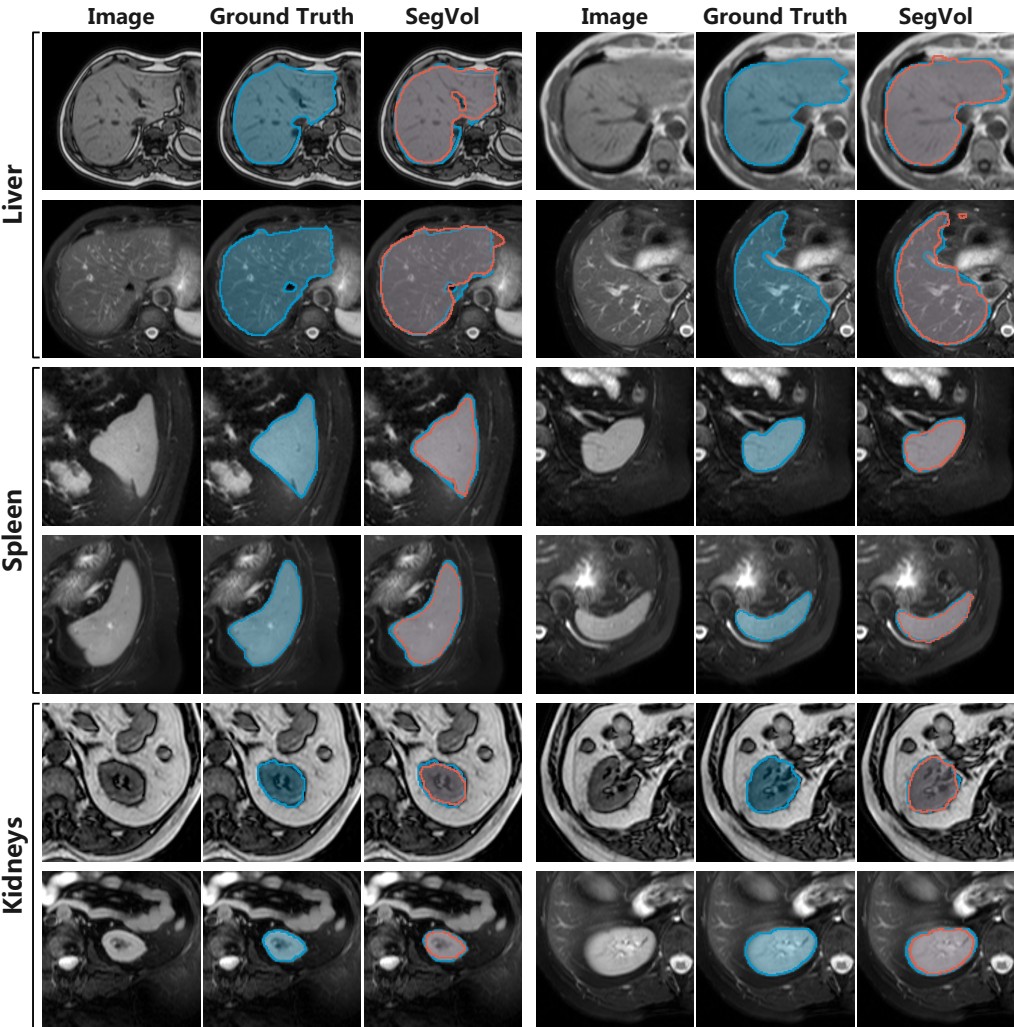

Figure 18: Visualized liver, spleen, and kidney prediction results of SegVol on 12 cases from MRI set of CHAOS[40, 41, 42]. For unseen MRI modality, SegVol is still able to segment these four organs relatively accurately.

# D   Evaluation Metrics

Each subset of the joint dataset is split into 80% training data and 20% test data. To ensure the absence of any data leaks, the hash value is utilized to compare the test set and training set. And in the comparative experiments, the model's parameters are all frozen.

We use the Dice Similarity Coefficient (Dice score) as a metric to evaluate the model, which is defined as $DSC = \frac{2|X \cap Y|}{|X|+|Y|}$. $|X \cap Y|$ is the cardinality of the intersection of the predicted segmentation sets $X$ and the ground truth sets $Y$. $|X|$ and $|Y|$ are the cardinalities of sets $X$ and $Y$ respectively. Dice

score is a commonly used metric for evaluating image segmentation tasks. It measures the degree of similarity between predicted segmentation and true segmentation, making it particularly suitable for evaluating the overlap degree of binary segmentation results.

# E    Additional Discussion

We present SegVol, a 3D foundational model for interactive and universal volumetric medical image segmentation. This method has been developed using 90K unlabeled CTs and 25 open-source medical datasets. This results in a universal segmentation tool capable of generating accurate responses for over 200 anatomical targets. Furthermore, SegVol demonstrates state-of-the-art volumetric segmentation performance when compared with both traditional task-specific methods[20, 21, 22, 23] and the recent SAM-like interactive methods[29, 38, 39, 28] in several comparative experiments. Despite its universality and high precision, SegVol maintains a simple architecture compared to other volumetric segmentation methods.

SegVol's capability of interactive and precise segmentation makes it a promising clinical aid tool. It can assist clinicians in identifying and quantifying tumor location, size, and shape changes within a patient's body[1] more accurately and rapidly. This precise monitoring aids clinicians in detecting tumor growth trends, assessing treatment effectiveness, and adjusting treatment plans as needed. Additionally, clinicians can use SegVol to accurately identify and segment important structures within a patient's body, such as organs, blood vessels, or the precise location of tumors and surrounding tissues, using high-resolution 3D images such as CT volumes. These precise segmentation results help clinicians better understand the patient's anatomical structures, plan surgical pathways, reduce surgical risks, and improve the accuracy and success rate of surgeries[3].

While SegVol is capable of understanding semantic-prompt composed of sentences, there remains a gap between it and the referring expression segmentation that involves complex semantic information and logical relationships. The establishment of a referring expression segmentation model needs more curated data with spatial annotations with text. Our SegVol provides a foundation for realizing referring segmentation of medical images, and we leave it as future work.

