# OpenReview forum: "SegVol: Universal and Interactive Volumetric Medical Image Segmentation"
_NeurIPS.cc/2024/Conference — NeurIPS 2024 spotlight_

### Official Review · Reviewer_Ykum · 2024-07-10

**Soundness:** 3
**Presentation:** 4
**Contribution:** 4
**Rating:** 8
**Confidence:** 4

**Summary:**

In this paper, the authors proposed SegVol, a 3D foundation segmentation model for medical images. This model supports universal and interactive volumetric medical image segmentation of more than 200 anatomical categories. This model is also well-designed with spatial and semantic prompts.

In the inference stage, the authors designed a zoom-out-zoom-in mechanism to enable efficient and precise segmentation results

Their extensive experiments show their method surpasses other SAM-like interactive segmentation methods.

**Strengths:**

Interesting concepts. The idea of integrating spatial and semantic prompts is interesting and achieves prospective results.

Paper clarity. The paper is overall well-written and structured. I also enjoyed the quality of the figures which help understanding the method.

Good results. The method achieves SoTA results compared with the SAM-like interactive segmentation methods.

**Weaknesses:**

I had a hard time finding weaknesses in the paper. Those I find are either nitpick or more directions for future work.

I put the weaknesses in the questions section.

**Questions:**

I would appreciate it if the authors could answer my questions.

1. Figure 2, the Violin plots seem confused and it is not cited in the paper. Can the authors explain more about this figure?

2. Will some of the 25 public volumetric medical segmentation datasets share the same data? Can the authors provide more detailed information about the datasets?

3. How do the authors process the collected data with inconsistent annotations? For example, different hospitals will have different annotations of the Aorta. Will this influence the results?

**Limitations:**

The authors adequately addressed their limitations in the manuscript as well as the appendix.

---

> ### Author Rebuttal · Authors · 2024-08-07
>
> Thank you for your valuable comments and kind words. We answer your questions as follows.
>
> > **Q1: Figure 2, the violin plots seem confused and it is not cited in the paper.**
>
> Figure 2 is cited on Page 7 Line 205: *‘We visualize the Dice score distributions of all methods in all the tasks as violin plots, depicted in Figure 2.’*
>
> Figure 2 on Page 5 and Table 2 on Page 6 describe the results of the same experiments from two different perspectives. Table 2 presents the precise quantitative results (the median value), while the violin plots in Figure 2 illustrate the distributions of the results for intuitive comparison.
>
> > **Q2: Will some of the 25 public volumetric medical segmentation datasets share the same data? Can the authors provide more detailed information about the datasets?**
>
> There are no duplicate cases in our processed 25 public datasets. We used the hash code to validate all cases (image-label pair) during the data processing phase. It was found that the LiTS dataset contains many duplicate cases which also exist in other datasets, thus the whole LiTS dataset was removed. Besides, one duplicate case was found and removed in the remaining 25 public datasets.
>
> For more detailed information about these open-source datasets, please refer to their official websites which are provided in Page 17 Table 5.
>
> > **Q3:** **How do the authors process the collected data with inconsistent annotations?**
>
> It is an important advantage of SegVol that the model is compatible with inconsistent but reasonable annotations. There is no need for additional data processing. For example, the whole ‘kidney’ is annotated in one hospital, while more specifically ‘left kidney’ and ‘right kidney’ are annotated in other hospitals. SegVol can learn to comprehend such inconsistent annotations and generalize well to users’ rough or precise semantic+spatial prompts in the testing phase.

---

> > ### Comment · Reviewer_Ykum · 2024-08-12
> >
> > I appreciate the author's detailed response. Most of my concerns have been addressed. I will maintain my original score.

---

> > > ### Author Response · Authors · 2024-08-13
> > >
> > > Thank you for your feedback. We will polish the final version based on your suggestions.

---

### Official Review · Reviewer_rczM · 2024-07-11

**Soundness:** 3
**Presentation:** 3
**Contribution:** 3
**Rating:** 7
**Confidence:** 5

**Summary:**

This paper proposes a 3D foundation segmentation model, named SegVol, supporting universal and interactive volumetric medical image segmentation. By scaling up training data to 90K unlabeled Computed Tomography (CT) volumes and 6K labeled CT volumes,  this foundation model supports the segmentation of over 200 anatomical categories using semantic and spatial prompts. Besides, a zoom-out-zoom-in mechanism is designed to facilitate efficient and precise inference on volumetric images.

**Strengths:**

(1) Collect and process 25 public volumetric medical segmentation datasets, encompassing over 200 anatomical categories. The pseudo label is introduced to relieve the spurious correlation in the training data.
(2) Implement massive 3D pre-training on 96K CT volumes and supervised fine-tuning on the  6k labeled datasets.
(3) Support spatial-prompt, semantic-prompt, and combined-prompt segmentation, achieving  high-precision segmentation and semantic disambiguation.
(4) Design a zoom-out-zoom-in mechanism that significantly reduces the computational cost, meanwhile preserving precise segmentation.

**Weaknesses:**

(1) As shown in Appendix Table 5, the numbers of trainset volumes is very dunbalanced， which will affect the performance of the proposed method. How the author address the above issue? They should give further analysis and validation.
(2) For unseen anatomical categories, what is the segmentation effect of SegVol? Provide relevant experiments for further explanation.
(3) What are the complexity、number of parameters and running speed of the SegVol, and provide a comparison with existing SOTA methods.
(4) Minor: The reference format should be unified，for example,[8]、[43]

**Questions:**

Answer the questions in Weakness Section

---

> ### Author Rebuttal · Authors · 2024-08-07
>
> Thank you for your constructive comments. Below we answer the specific questions.
>
> > **Q1: As shown in Appendix Table 5, the number of train-set volumes is very unbalanced, which will affect the performance of the proposed method.**
>
> It is supposed that the unbalanced training set may cause poor performance on the minor categories/domains in some tasks. However, this problem does not matter in our work. We studied this problem by training SegVol on those small sub-datasets and comparing its performance to that trained on the joint of all datasets. Specifically, BTCV, MSD-spleen, and FLARE22 are used as the minor sub-datasets (domains), which have 24, 32, and 40 training samples respectively, and account for only 0.5%, 0.7%, and 0.9% of the whole training set. The following table shows that SegVol trained on the whole dataset, which consists of 25 sub-datasets, achieves much better performances than those trained on the small sub-datasets respectively. The results indicate that SegVol, as a foundation model, performs well on minor sub-datasets (domains) and does not suffer from the problem of unbalanced training data. We will provide more analysis in the final version.
>
> | **Avg. Dice Score**                 |    BTCV    | MSD-spleen |  FLARE22   |
> | :---------------------------------- | :--------: | :--------: | :--------: |
> | SegVol trained on small sub-dataset |   0.5195   |   0.9471   |   0.5567   |
> | SegVol trained on the whole dataset | **0.8058** | **0.9597** | **0.8822** |
>
> > **Q2: For unseen anatomical categories, what is the segmentation effect of SegVol?**
>
> In Page 6 Table 2 and Page 7 Line 202, we presented the experiments on ULS23, where most lesion categories are unseen for SegVol. The unseen categories include most abdominal lesions, all bone lesions, some lung lesions, and all mediastinal lesions. For unseen categories, we use spatial prompts and general semantic prompts, e.g., ‘lesion’ or ‘tumor’, to drive the model. The results show that SegVol can achieve good segmentation performance (Dice: 0.7046) on these unseen categories.
>
> After the review phase, we will host an online running model for users to test various cases from unseen categories.
>
> > **Q3: What are the complexity, number of parameters and running speed of the SegVol?**
>
> The following table shows the Total Parameters, the average Multiply-Accumulates(MACs), and the average Time required to process a case of the different methods. The comparison indicates that our method takes less computational cost while achieving much better performance.
>
> Note that when calculating MACs Per Case and Time Per Case, the slice-by-slice calculation of the 2D method and the scanning process of the 3D method are accumulated respectively. SAM-MED3D only processes volume with a size of $128\times128\times128$. The experiments are implemented on the validation set of AMOS22, and the setting is the same as that in Sec. 3.2.
>
> | **Method** | **Total Parameters** | **Avg. MACs Per Case↓** | **Avg. Time Per Case (s)↓** | **Avg. Dice Score↑** |
> | :--------- | :------------------: | :---------------------: | :-------------------------: | :------------------: |
> | SAM        |         94M          |         1.3e+13         |           2.1764            |        0.5271        |
> | MedSAM     |         94M          |         1.3e+13         |           2.1886            |        0.5371        |
> | SAM-MED2D  |         271M         |         2.3e+12         |           3.5547            |    *0.6668*     |
> | SAM-MED3D  |         101M         |       **1.0e+11**       |         **0.1768**          |        0.6200        |
> | SegVol     |         181M         |     *6.7e+11*      |        *0.3283*        |      **0.8593**      |
>
>
> > **Q4: Minor: The reference format should be unified.**
>
> Thanks for your reminder. We will unify the reference format in the final version.

---

> ### Author Response · Authors · 2024-08-12
>
> Dear Reviewer rczM,
>
> We hope our response addresses your questions. Please feel free to reach out if you have any further inquiries. We would greatly appreciate it if you could consider raising your rating.
>
> Thank you very much!

---

### Official Review · Reviewer_8irr · 2024-07-24

**Soundness:** 4
**Presentation:** 2
**Contribution:** 4
**Rating:** 8
**Confidence:** 4

**Summary:**

The paper describes the utilization of SegVol, a deep learning model, to segment any organ/tumor/lesion on 3D CT data.

**objectives**
Create a universal segmentation model that can segment with :
-  any type of labeling
-  good performances on complex tasks
-  low computational cost (i.e., no sliding window)

**contributions**
The authors contribute to the state of the art in 3 ways :
- first model for segmenting over 200 anatomical categories
- support spatial (points/bbox) and semantic prompt (text describing each category)
- introduction of a zoom-out-zoom-in mechanism to improve the model's efficiency and accuracy

**datasets**
- 25 open-source segmentation CT datasets + background filtering
- generation of pseudo-labels with Felzenswalb-Huttenlocher to remove spurious correlation between the datasets

**architecture**
The whole architectures is divided into four blocks:
- image encoder: 3D ViT pretrained using SimMIM on 96K unlabeled CTs + fine-tuning on 6K labeled CTs
- text encoder: the CLIP model is used to generate a text embedding from the text prompt. The text prompt only consists in the name of the anatomical region to be segmented.
- prompt encoder: text embedding and spatial prompt (bbox+points) are given to generate the prompt embedding.
- mask decoder: from the prompt embedding, the segmentation is generated.

**training algorithm**
- the encoder is pretrained using SimMIM
- the model is fine-tuned using an algorithm described in appendix B. For each epoch, an iteration of training is made using the ground truth, and a second one using the pseudo masks.

**results**
Experimental setup is given, as well as a description of the three external datasets used for the evaluation.
- comparison with SAM-like methods : Table 2 shows the superiority of the method with Dice score > 0.7 on each task. Figure 2 helps to visualize the dice score distribution for each model, on each task. An additional comparison to standard fully supervised segmentation models such as nnUNet, SwinUNetR is provided. Superiority of SegVol is also shown compared to this model on Figure 8 and Table 6.
- ablation studies
  - Zoom-out-zoom-in mechanism : Table 3 shows the interest of the method, improving performances and reducing duration compared to sliding window
  - Scaling up training data : Fig 3.(a) shows that the model performances improve as the number of training data increases.
  - Prompt impact : Fig 3 (a and b) shows that point prompt is inferior to text prompt, and best prompt combination is bbox+text
- Case studies
  - Disambiguation via semantic-prompt: Figure 4 shows that when the spatial prompt is not sufficient to describe the segmenting task, the text prompt adds the complementary information to locate the good anatomical region.
  - spatial-prompt impact: Figure 5 shows how the points prompt are related to the results. When choosing two points, the category becomes clear, whereas with one point can identify two categories.

**discussion**
- Scalability : the authors believe that the model have a strong model scalability.
- Generalizability : Generalization of SegVol to MRI is discussed, with an example given in appendix C showing a strong dice score of 80%.
- Limitations : only limitation is that text prompt only consider a category, and not a full sentence with logical reasoning.
- Broader impact : the authors believe that their method is universal and doesn't have any negative impact.

**Strengths:**

- **originality** : the concept of SegVol is not original itself. However, the addition of the zoom-in-zoom-out principle and the combination of bbox + text + points prompts is new in this kind of models.
- **quality** : the work is done rigorously, respecting a scientific reasoning all along the article.
- **clarity** : the paper lacks clarity as there are two parts difficult to understand.
  - 1) The description of the training algorithm is not made in the text, only the steps are written in appendix B.
  - 2) The hyperparameters regarding the model's dimension (number of layer, ViT dimension) is not explicited.
- **significance** : The results are impactful for the community of medical image segmentation, and could have plenty of applications. The property of segmenting multiple organs, from multiple entirely different datasets is remarkable.

**Weaknesses:**

The only weakness regards the clarity of the training algorithm, which is hard to understand even with appendix B and training steps.
This point could be improved by describing the steps in the appendix with a paragraph, or maybe with another diagram that follows the exact same steps.

**Questions:**

**datasets**
l. 83 : Why is Felzenswalb-Huttenlocher (FH) algorithm to generate pseudo masks ?
l. 81 : How is it supposed to remove spurious correlation between datasets ?

**architecture**
l. 96 : The architecture principle is described, but some details are lacking such as the number of layers of each network, the number of heads in the ViT, etc.
l. 133 : why is the computational cost reduced using the zoom-in-zoom-out principle compared to sliding window? clarify this point.
l. 140 : how are prompts generated from the coarse segmentation masks? clarify if the prompts are generated thanks to the prompt encoder as a bbox, or points.

**results**
l. 169 : Could you indicate the training time (SimMIM and finetuning) of the model?
l. 169 : Why did you choose these 3 specific datasets as external test sets?

**discussion**
l. 257 : Why is data salability a good thing ? Why is this part in the discussion? The data scalability of the model is not discussed or compared with other models. If not discussed, move to the conclusion.

**Limitations:**

Only one limitation is addressed, considering the text prompt which cannot be a logical sentence at the moment.
However, the amount of data required to train the model is not discussed. Would a few-shot finetuning method work on small datasets (10 samples)?

---

> ### Author Rebuttal · Authors · 2024-08-07
>
> Thank you for your high recognition of our paper and the detailed feedback. We answer your questions as follows.
>
> > **Q1: Clarity of the training algorithm. Describing the steps with a paragraph or diagram.**
>
> Thank you for the suggestion. We will provide the detailed text description of the Training Algorithm in the final version. Here, we give a simple diagram, Figure 1 in the attached PDF, and a brief description for your reference. Specifically, each case (training sample) consists of an Image *x*, a Ground Truth(GT) Mask Set *Y*, and a Pseudo Mask Set *Z*. The training loss of each sample consists of the ground-truth loss and the pseudo loss. The ground-truth loss is computed by inputting the image, the ground-truth mask (label) and the sampled prompt into the model, while the pseudo loss is computed by inputting the image, the pseudo label and the fixed prompt into the model. Finally, the model is updated by minimizing the weighted sum of the two losses.
>
>
> > **Q2: Why is FH algorithm to generate pseudo masks? How is it supposed to remove spurious correlation between datasets?**
>
> The FH algorithm is an unsupervised segmentation method that can generate segmentation masks purely based on the voxel similarity of regions in CT scans. In the beginning, we also tried several candidates, including *morphological_chan_vese*, *morphological_geodesic_active_contour*, *quickshift*, *slic*, etc. Finally, we found the FH algorithm performs best, and thus chose it to generate pseudo masks.
>
> Since in most datasets, the 3D CTs are annotated with only a few segmentation categories, which causes a spurious correlation between the labeled categories and the specific dataset. Using pseudo masks can supplement unlabeled categories in a dataset, therefore relieving the spurious correlation.
>
> > **Q3: Some details are lacking such as the number of layers of each network, the number of heads in the ViT, etc.**
>
> Thanks for reminding, we will add the network details in the paper. Here are main hyper-parameters of the ViT in SegVol:
>
> patch_size=(4, 16, 16), num_layers=12, num_heads=12, hidden_size=768.
>
> The code and model weights of SegVol will be released after the review period.
>
> > **Q4: Why is the computational cost reduced using the zoom-out-zoom-in principle compared to the sliding window?**
>
> The traditional sliding window method requires scanning the entire 3D-CT, processing *hundreds* of windows. In contrast, the proposed zoom-out-zoom-in mechanism only requires one global inference of 3D-CT and scanning the ROI with *dozens* of windows.
>
> > **Q5: How are prompts generated from the coarse segmentation masks?**
>
> The process of prompts generated from the coarse segmentation masks is the same as that of prompts generated from ground truth masks, where point prompts are created in or beside the targets, bbox prompts are formed near the target boundaries, and text prompts are derived from category names. The detailed strategies are presented in Page 4 Sec 2.3.
>
> > **Q6: Could you indicate the training time (SimMIM and finetuning)?**
>
> SimMIM pre-training time: ~20$\times$8 GPU hours on NVIDIA A100-SXM4-40GB.
>
> Finetuning time: ~300$\times$8 GPU hours on NVIDIA A100-SXM4-40GB.
>
> > **Q7:** **Why did you choose these 3 specific datasets as external test sets?**
>
> AMOS22 is a popular dataset for medical image segmentation. Many works are trained and evaluated on it. The validation set of AMOS22, as a representative dataset of abdominal organs, has a large number of samples and includes 15 major abdominal organs.
>
> The ULS23 dataset is a novel large-scale lesion segmentation dataset with thousands of cases covering various lesions, which is a challenging benchmark of lesion segmentation.
>
> SegTHOR dataset focuses on the thorax, which can be a supplement of AMOS22 (abdomen) and ULS23 (lesions). On the whole, the three datasets cover most of the segmentation tasks of organs and lesions in the thorax and abdomen.
>
> > **Q8: Why is data salability a good thing? Why is this part in the discussion?**
>
> The scaling law of foundation models has been verified in multiple CV and NLP tasks. We achieved the success of scaling law in 3D medical segmentation by the design of universal prompts and pseudo masks for joint learning on datasets with inconsistent annotations. We provided a preliminary experiment of the data scaling law in Figure 3 (a) Page 7, which shows that 1) the performance improves significantly with more training data, 2) and SegVol has not yet reached its ceiling if more training data is provided. We will provide more detailed discussion and comparison to other methods in the final version.
>
> > **Q9: The amount of data required to train the model (SegVol) is not discussed. Would a few-shot finetuning method work on small datasets (10 samples)?**
>
> Figure 3 (a) in Page 7 demonstrates the relationship between the model performance and the amount of finetuning training data. Generally speaking, more training data leads to better performance.
>
> We conducted the few-shot finetuning experiment on small datasets, FLARE22 (40 training cases) and MSD-spleen (32 training cases), to study the few-shot learning ability of SegVol. The table below demonstrates that 1) finetuning SegVol on dozens of samples works well on easy datasets such as MSD-spleen, in which the few-shot performance is close to the joint finetuning on all datasets; 2) for challenging datasets such as FLARE22, finetuning on all datasets can achieves much better performance than few-shot finetuning.
>
> | Avg. Dice Score | 100  epochs | 200 epochs | 300 epochs | 400 epochs | 500 epochs | SegVol* |
> | :-------------- | :---------: | :--------: | :--------: | :--------: | :--------: | :-----: |
> | FLARE22         |   0.0463    |   0.4028   |   0.4926   |   0.5617   |   0.5567   | 0.8822  |
> | MSD-spleen      |   0.7566    |   0.7866   |   0.9433   |   0.9454   |   0.9471   | 0.9597  |
>
> *Note: **SegVol\*** represents the model finetuned on all datasets.*

---

> ### Comment · Reviewer_8irr · 2024-08-13
>
> Thank you for your detailed answers to each of my interrogations.
> All the questions have been answered by the authors.
>
> Q1 : the algorithm is clearer with this small text, it should be included in the article.
> Q2 : the choice of FH is then arbitrary, it should be specified in the article.
> Q3 : For the network details, maybe a results of a torchinfo summary, or something in this idea could be added in appendix so that every hyperparameters could be seen visually (it would be be a bonus, but optional).
> Q9 : The results of joint finetuning compared to few-shot finetuning is relevent, and could be at least mentionned.
>
> I will maintain my grading, considering the clarity (presentation) will be improved, but is not yet done.

---

> > ### Author Response · Authors · 2024-08-13
> >
> > Thank you for the further feedback. We will polish the final version based on your suggestions.

---

### Author Rebuttal · Authors · 2024-08-07

We sincerely appreciate the reviewers’ constructive comments and valuable suggestions.

We are glad to see that our paper received high praise from all reviewers. Especially, **Reviewer 8irr** finds that ‘*the results are impactful for the community of medical image segmentation, and could have plenty of applications.*’ **Reviewer Ykum** thinks that ‘*I had a hard time finding weaknesses in the paper. Those I find are either nitpick or more directions for future work.*’

We address the specific questions from reviewers as follows.

---

### Decision · Program_Chairs · 2024-09-25

**Decision:**

Accept (spotlight)

**Comment:**

The paper introduces SegVol, a universal deep learning model designed for segmenting any organ, tumor, or lesion in 3D CT data. SegVol leverages a substantial amount of training data and incorporates both spatial and semantic prompts to enhance segmentation accuracy. The model is noted for its innovative zoom-out-zoom-in mechanism, which improves efficiency and performance without the need for a sliding window approach.

Strengths :
While SegVol itself is not entirely novel, its combination of zoom-in-zoom-out principles and prompt types is innovative.
The work is rigorous and scientifically sound.
The paper is generally clear, though it lacks detailed descriptions of the training algorithm and model hyperparameters.
The model's ability to segment multiple organs from diverse datasets is highly impactful.

Weaknesses :
The training steps are hard to understand and could benefit from additional explanation and diagrams.
The unbalanced volume of training data could affect performance.
The segmentation performance on unseen categories needs further validation.
The complexity, parameter count, and running speed of SegVol need to be compared with existing SOTA methods.

Recommendations :
The training steps could be better presented even with Appendix B
Only one limitation is mentioned regarding the text prompt's current capacity.
The discussion of the required amount of training data required could be addressed, and the potential for few-shot fine-tuning could be explored as well.